# Sirtuin 6 inhibition protects against glucocorticoid-induced skeletal muscle atrophy by regulating IGF/PI3K/AKT signaling

Chronic activation of stress hormones such as glucocorticoids leads to skeletal muscle wasting in mammals. However, the molecular events that mediate glucocorticoid-induced muscle wasting are not well understood. Here, we show that SIRT6, a chromatin-associated deacetylase indirectly regulates glucocorticoid-induced muscle wasting by modulating IGF/PI3K/AKT signaling. Our results show that SIRT6 levels are increased during glucocorticoid-induced reduction of myotube size and during skeletal muscle atrophy in mice. Notably, overexpression of SIRT6 spontaneously decreases the size of primary myotubes in a cell-autonomous manner. On the other hand, SIRT6 depletion increases the diameter of myotubes and protects them against glucocorticoid-induced reduction in myotube size, which is associated with enhanced protein synthesis and repression of atrogenes. In line with this, we find that muscle-specific SIRT6 deficient mice are resistant to glucocorticoid-induced muscle wasting. Mechanistically, we find that SIRT6 deficiency hyperactivates IGF/PI3K/AKT signaling through c-Jun transcription factor-mediated increase in IGF2 expression. The increased activation, in turn, leads to nuclear exclusion and transcriptional repression of the FoxO transcription factor, a key activator of muscle atrophy. Further, we find that pharmacological inhibition of SIRT6 protects against glucocorticoid-induced muscle wasting in mice by regulating IGF/PI3K/AKT signaling implicating the role of SIRT6 in glucocorticoid-induced muscle atrophy.

Maintenance of skeletal muscle mass and function is crucial for the healthy life span of humans. The skeletal muscle mass is maintained by a critical balance between the protein synthesis and the degradation pathways. However, this fine balance is lost in conditions of chronic stress such as in various aging-related diseases like diabetes, cancer, renal failure, cardiac failure, AIDS, and chronic obstructive pulmonary disease[1–3]. Notably, a major contributor to chronic stress-induced muscle wasting is elevated serum levels of glucocorticoids[4,5]. In the muscle cells, glucocorticoids bind to the glucocorticoid receptor (GR), which induces its nuclear translocation and recruitment to the glucocorticoid response element (GRE). The binding of GR to GRE then alters the expression of multiple transcription factors which control the muscle atrophy program[4]. Of these, the FoxO or forkhead box

transcription factors play a major role by inducing the expression of atrogenes which include specialized E3 ubiquitin ligases such as Atrogin-1/MAFbx (muscle atrophy F-box protein) and MuRF-1 (muscle RING finger-1 protein)[6,7]. Apart from increasing the rate of muscle protein degradation, glucocorticoids also inhibit protein synthesis by inhibiting the stimulatory action of insulin or IGF (Insulin-like Growth Factor) on muscle protein synthesis[8]. Although glucocorticoid-induced muscle atrophy has been extensively studied, the molecular regulator(s) capable of modulating both muscle atrophy and hypertrophy are still not well understood.

Extensive studies indicate that exercise and dietary restriction could reverse muscle atrophy[9–13]. Notably, recent studies indicate that Sirtuins, a family of NAD+ dependent deacetylases, are stress-

✉ e-mail: rmostoslavsky@mgh.harvard.edu; rsundaresan@iisc.ac.in

responsive enzymes and are involved in regulating the molecular changes occurring during exercise and dietary restriction[14]. Seven sirtuin isoforms (SIRT1-7) have been identified in mammals, which have distinct subcellular localizations. Of the seven isoforms, Sirtuin 6 (SIRT6) is a chromatin-associated NAD$^+$-dependent enzyme with deacetylase, deacylase, and mono-ADP-ribosylase activities[15]. SIRT6 targets both histone and non-histone proteins to control diverse cellular processes including transcription, DNA damage response, heterochromatin silencing, regulation of telomere stability, and cellular senescence[16]. Notably, SIRT6 deficient mice undergo accelerated aging and suffer from severe complications including severe hypoglycemia, loss of subcutaneous fat, a curved spine, and lymphopenia[17]. On the other hand, SIRT6 overexpression protects mice from aging-related complications[18] and also extends the life span of male mice via inhibition of IGF signaling[19]. Studies indicate that SIRT6 is a key regulator of many aging-related diseases such as heart disease, diabetes, obesity, cancer, metabolism, and inflammation[16]. Our previous findings suggest that SIRT6 levels decrease in heart failure patients and this downregulation is linked to the development of cardiac hypertrophy and heart failure in mice[20,21]. Although SIRT6 plays a major role in aging and aging-related diseases, its role in skeletal muscle atrophy, a key aging-related pathology, is not clearly understood.

In the present work, we investigated the role of SIRT6 in the maintenance of skeletal muscle homeostasis and glucocorticoid-induced muscle atrophy using a muscle-specific SIRT6 knockout mice. Interestingly, we find that SIRT6 deficiency confers protection against glucocorticoid-induced muscle wasting through increased activation of PI3K/AKT signaling.

## Results

### SIRT6 levels increase during glucocorticoid-induced muscle atrophy

To study the role of SIRT6 in glucocorticoid-induced muscle atrophy, we used dexamethasone (Dex, a glucocorticoid analog) to induce muscle atrophy in mice, which is a well-established mouse model for muscle wasting. Dex administration for a period of 7 days significantly reduced the bodyweight of animals, which was further reduced 15 days after Dex administration (Fig. 1a and Supplementary Fig. 1A). Notably, Dex treatment resulted in significant reduction of the muscle weight to tibia length ratio (MW/TL) and absolute muscle mass of the tibialis anterior (Fig. 1b and Supplementary Fig. 1B), gastrocnemius (Supplementary Fig. 2A and B), and soleus (Supplementary Fig. 3A and B) muscles, but not of quadriceps (Supplementary Fig. 4A and B), triceps (Supplementary Fig. 5A and B), and biceps (Supplementary Fig. 6A and B) muscles. Similarly, Dex administration significantly reduced mean myofiber cross-sectional area (CSA) in tibialis anterior (Fig. 1c, d and Supplementary Fig. 1C), gastrocnemius (Supplementary Fig. 2C and D), and soleus (Supplementary Fig. 3C and D) muscle, but not in quadriceps (Supplementary Fig. 4C and D), triceps (Supplementary Fig. 5C and D), or biceps (Supplementary Fig. 6C and D) muscles. Consistent with this, we observed that Dex administration increased the frequency of smaller-sized myofibers in tibialis anterior (Fig. 1c, e, and Supplementary Fig. 1C), gastrocnemius (Supplementary Fig. 2C, E), and soleus (Supplementary Fig. 3C, E) muscle, while the size distribution of myofibers in other muscle types were unaffected by Dex administration (Supplementary Figs. 4C, E, 5C, E, 6C, E). These data confirm the Dex-induced reduction of muscle mass consistent with previous reports[22,23].

We next performed myosin heavy chain (MHC) staining to assess the fiber type composition of the different muscles in vehicle and Dex-treated mice. Our analysis shows that tibialis anterior, gastrocnemius, quadriceps, triceps, and biceps muscles are composed majorly of fast-type fibers such as MHC IIA, MHC IIB, and MHC IIX fibers (Supplementary Figs. 1C, D, 2C, F, 4C, F, 5C, F, 6C, F)

while soleus muscle is comprised of ~ 40% of slow type fiber namely the MHC1 (Supplementary Fig. 3C, F) as shown previously[24,25]. Notably, Dex administration induced an increase in slow or MHC I fiber type in all muscle types with a concomitant reduction in different fast fiber types in different muscles (MHC IIB and MHC IIX in tibialis anterior and soleus muscle, and MHC IIB in gastrocnemius, and triceps muscles) (Supplementary Figs. 1C and D, 2C, F, 3C, F, 4C, F, 5C, F, 6C, F) as shown previously[26,27]. In addition, we also found increased expression of atrophy markers such as Atrogin-1, MuRF-1, FoxO1, and FoxO3 in Dex-treated gastrocnemius muscle samples, confirming the induction of muscle atrophy (Fig. 1f). These data confirm that we could successfully recapitulate Dex-induced muscle atrophy model as reported previously[22–25].

In this Dex-induced muscle atrophy model, we next tested the expression level of SIRT6 using qPCR and western blotting analysis. Interestingly, we observed that Dex administration for 24 h increased the mRNA levels of SIRT6 in the gastrocnemius muscle (Fig. 1g). Moreover, Dex administration for 24 h increased the protein levels of SIRT6 in the tibialis anterior, gastrocnemius, and soleus muscles (Fig. 1h and Supplementary Fig. 7A and B), the same muscle types which show significant loss of muscle mass upon chronic Dex administration (Fig. 1b and Supplementary Figs. 1B, 2A and B, 3A and B). Notably, the SIRT6 protein levels continued to remain high after 7 and 15 days of Dex administration only in the tibialis anterior muscle (Fig. 1h), but not in other muscle types tested (Supplementary Fig. 7A–E). However, SIRT6 levels reduced significantly after 7 days and 15 days of Dex administration in gastrocnemius muscle (Supplementary Fig. 7A). In contrast to these muscle types, the SIRT6 levels were unchanged in quadriceps, triceps, and biceps muscles where the muscle mass also was not significantly changed by Dex administration (Supplementary Fig. 7C–E and Supplementary Figs. 4A and B, 5A and B, 6A and B). Taken together, these results suggest that SIRT6 is specifically upregulated in the muscle types that show significant muscle loss upon Dex administration.

To recapitulate in vitro the role of SIRT6 in glucocorticoid-induced muscle atrophy, we cultured murine primary myotubes and treated them with Dex. Our results show that Dex treatment markedly reduced the diameter of myotubes (Fig. 1i, j). Consistent with previous studies[28,29], we find that Dex treatment markedly reduces global protein synthesis in primary myotubes (Fig. 1k). These observations indicate the onset of atrophic changes in myotubes upon Dex treatment. Interestingly, similar to our in vivo findings, we observed increased levels of both mRNA and protein of SIRT6 in 24 h Dex-treated myotubes (Fig. 1l, m). Moreover, protein levels of SIRT6 decreased in 48 h of Dex-treated myotubes (Data shown in Fig. 8c). These findings indicate that increased SIRT6 levels may be associated with skeletal muscle atrophy.

### SIRT6 overexpression reduces myotube diameter in a cell-autonomous manner

To understand the significance of increased levels of SIRT6 during muscle atrophy, we overexpressed SIRT6 in primary myotubes and assessed the myotube diameter. Interestingly, we find that SIRT6 overexpression significantly reduced the diameter of myotubes (Fig. 2a–c), indicating that SIRT6 overexpression is sufficient to reduce myotube diameter. Moreover, we observed increased expression of MuRF-1/2/3 and Atrogin-1 in SIRT6 overexpressing myotubes (Fig. 2d, e). We confirmed the specificity of the MuRF-1/2/3 and Atrogin-1 antibodies by observing their specific reduction in MuRF-1 and Atrogin-1 depleted myotubes respectively (Supplementary Fig. 8A and B). SIRT6 overexpression was also sufficient to repress global protein synthesis in primary myotubes (Fig. 2f). Overall, these findings suggest that SIRT6 overexpression spontaneously induces expression of atrophy markers, reduces protein synthesis, and reduces myotube size in a cell-autonomous manner.

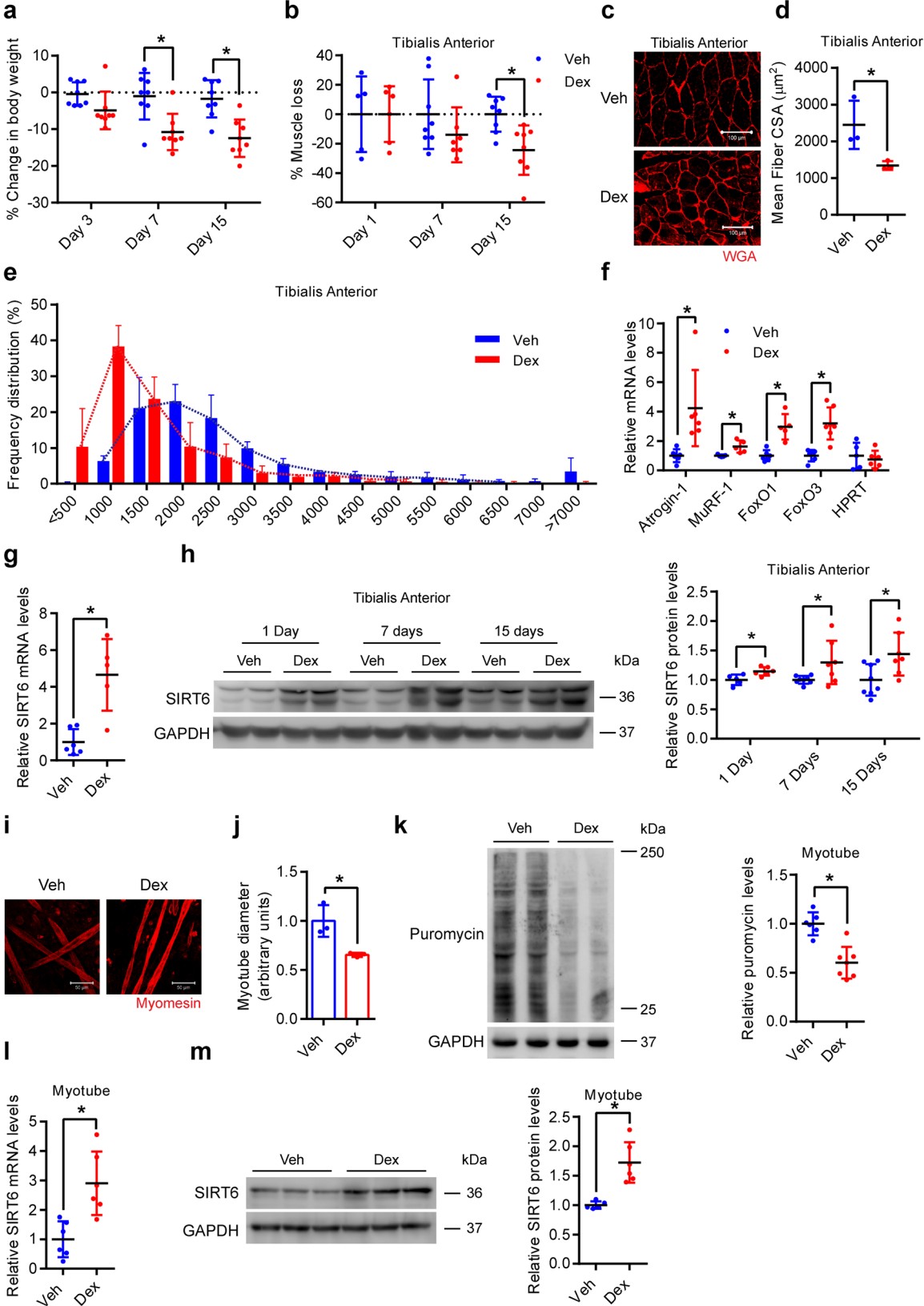

## SIRT6 depletion attenuates the dexamethasone-induced reduction in myotube size

Since SIRT6 overexpression reduces myotube diameter, we tested whether SIRT6 depletion could protect against Dex-induced reduction in myotube diameter. Notably, we found that SIRT6 depletion significantly increases myotube diameter at the basal level (Fig. 3a–c). In

addition to quantifying myotubes from different fields, we also acquired a tile scan image of a quarter of a coverslip to obtain an unbiased view of all myotubes present in the field from both groups. We indeed find that the myotubes were appreciably thicker when SIRT6 was depleted (Supplementary Fig. 9A). Moreover, SIRT6-depleted myotubes did not show any significant reduction in their

**Fig. 1 | Dexamethasone increases SIRT6 levels in muscles and myotubes.**
**a** Change in body weight after Dex administration. $n = 8$. **b** Loss in tibialis anterior (TA) muscle after Dex administration. Dex-induced % muscle loss is shown relative to Veh from each time-point. Day 1-Veh $n = 5$, Day 1-Dex $n = 6$, Day 7-Veh, Day 15-Veh and Day 15-Dex $n = 8$, Day 7-Dex $n = 7$. **c** Representative images showing wheat germ agglutinin (WGA; red) stained Dex-administered mice TA. Scale bar = 100 μm. **d** Mean fiber cross-sectional area (CSA) in mice described in **c**. $n = 3$. **e** Frequency distribution of fiber CSA in mice described in **c**. $n = 3$. **f** qPCR analysis for relative expression of atrophy genes in Dex injected mice gastrocnemius. HPRT was used as negative control. Atrogin-1, FoxO1-Veh, FoxO3-Veh, HPRT-Dex $n = 6$, MuRF-1, FoxO1-Dex, HPRT-Veh $n = 5$, FoxO3-Dex $n = 7$. **g** qPCR analysis for relative expression of SIRT6 in 24 h Dex injected mice gastrocnemius. Veh $n = 6$, Dex $n = 5$. **h** Western blot (left) and quantification (right) of SIRT6 in Dex-administered mice

TA. Day 1-Veh $n = 5$, Day 1-Dex $n = 6$, Day 7-Veh, Day 15-Veh $n = 8$, Day 7-Dex, Day 15-Dex $n = 7$. **i** Representative confocal image of Dex-treated myotubes stained with myomesin (red). Scale bar = 50 μm. **j** Quantification of fiber diameter in myotubes described in **i**. $n = 3$. **k** Western blot (left) and quantification (right) of SUnSET analysis in Dex-treated myotubes. Veh $n = 6$, Dex $n = 7$. **l** qPCR analysis for relative expression of SIRT6 in Dex-treated myotubes. $n = 6$. **m** Western blot (left) and quantification (right) of SIRT6 in Dex-treated myotubes. Veh $n = 5$, Dex $n = 6$. Data presented as mean ± s.d., *$p < 0.05$. Two-way ANOVA with Bonferroni post hoc test was used for statistical analysis (**a**, **b**) and two-tailed unpaired Student's $t$-test (**d**, **f**, **g**, **h**, **j**–**m**). Dex (10 mg/kg/day) was administered (**a**–**h**). Myotubes were treated with Dex for 48 h (**i**–**k**), 24 h (**l**, **m**). Source data are provided as a Source Data file.

diameter upon Dex treatment, contrary to the control myotubes (Fig. 3a–c). Consistent with this, control, but not SIRT6 depleted myotubes, treated with Dex exhibited significantly increased levels of MuRF1/2/3 and Atrogin-1 (Fig. 3d, e). Similarly, SIRT6 depletion in myotubes prevented the decrease in protein synthesis following Dex treatment, when compared to control myotubes (Fig. 3F). Collectively, these results suggest that SIRT6 depletion attenuates Dex-induced reduction in myotube size.

## Muscle-specific SIRT6 deficient mice are resistant to Dex-induced muscle atrophy

To further understand the in vivo significance of the role of SIRT6 in the regulation of skeletal muscle homeostasis, we generated a muscle-specific SIRT6 deficient mouse (msSIRT6-KO). msSIRT6-KO mice were obtained by crossing SIRT6-fl/fl mice with mice expressing Cre recombinase under the control of human alpha-skeletal actin promoter (ACTA) (Fig. 4a). Our western blotting analysis confirmed the deletion of SIRT6 in different muscle types (Fig. 4a and Supplementary Fig. 10A). Further, western blotting analysis showed the specific deletion of SIRT6 in the skeletal muscles but not in other tissues such as kidney or liver (Supplementary Fig. 10B). Nevertheless, a thin band was observed sometimes in the KO muscle lysates, which could be due to SIRT6 from non-muscle cells such as blood cells or immune cells (Fig. 4a and Supplementary Fig. 10B). We next analyzed the mRNA and protein levels of the various Sirtuins in the skeletal muscle of msSIRT6KO mice. Our qPCR analysis suggested that SIRT6 deletion in muscle increased SIRT3 mRNA levels and decreased SIRT6 levels while the levels of other Sirtuin isoforms were unaltered (Supplementary Fig. 10C). Moreover, muscle-specific SIRT6 deletion led to increased protein levels of SIRT2 and SIRT4 (Supplementary Fig. 10D). The reasons for these changes are however unclear and further studies are needed to understand the significance of these compensatory changes of different Sirtuin isoforms in muscle.

The msSIRT6-KO mice were viable, fertile, and showed no major physiological defects until one year of age. The mice did not show any marked changes in biochemical parameters such as fasting blood glucose, serum triglyceride, and total serum protein levels when compared to controls (Supplementary Fig. 11A–C). In addition, we measured the levels of free amino acids in msSIRT6-KO mice muscles by high-performance liquid chromatography (HPLC). Our results indicate that, with the exception of reduced alanine, the levels of amino acids are not different in the muscles of control and msSIRT6-KO mice (Supplementary Fig. 11D), suggesting that SIRT6 deficiency may not cause any major defect in amino acid biosynthesis or catabolism. We also performed qPCR analysis for genes involved in metabolism, energy expenditure, and stress response. We found that the expression of PGC1-α, a master regulator of mitochondrial biogenesis, and genes involved in mitochondrial respiration such as UCP2, SDH, COX4I1, NRF2, and ATP5A1 were not affected by SIRT6 deficiency in muscle (Supplementary Fig. 11E). Further, our analysis of body weights indicated that msSIRT6-KO mice show bodyweight similar to

controls (Supplementary Fig. 11F). In addition, there was no significant difference in the muscle weight to tibia length ratio of quadriceps and gastrocnemius muscles between the msSIRT6-KO mice and the controls, at the basal level (Supplementary Fig. 11G). Taken together, our comprehensive characterization of the msSIRT6-KO mice show that these mice do not suffer from any major abnormalities under basal conditions.

Since our in vitro findings suggest that SIRT6 depletion protects myotubes against Dex-induced reduction in size, we tested whether msSIRT6-KO mice are resistant to Dex-induced skeletal muscle wasting. We chronically administered Dex to msSIRT6-KO mice for 15 days and studied the muscle phenotype. While Dex-treated control mice display drastically reduced bodyweight, msSIRT6-KO do not show any significant weight loss upon Dex treatment (Fig. 4b and Supplementary Fig. 12A). Similarly, we observed a significant reduction in muscle mass of tibialis anterior (Fig. 4c and Supplementary Fig. 12B) and gastrocnemius muscle (Supplementary Fig. 12C and D) of Dex-treated control mice, while msSIRT6-KO mice were resistant to Dex-induced atrophy (Fig. 4c and Supplementary Fig. 12B–D). However, Dex treatment did not cause any significant change in muscle weights of the quadriceps, triceps, or biceps (Supplementary Fig. 12E–J). We further tested whether the observed phenotypic changes in Dex-treated mice accompany biochemical changes in muscle. We performed Bradford assay to test the changes in the levels of soluble protein concentration in control and msSIRT6-KO mice at the basal level and upon Dex treatment. Our result showed that soluble protein concentration in msSIRT6-KO mice were comparable to controls at the basal level and upon Dex administration (Supplementary Fig. 12K). However, Dex-treated control mice showed a trend toward decrease in soluble protein concentration, although the changes were not statistically significant (Supplementary Fig. 12K). These results indicate that Dex-induced phenotypic changes in control mice are independent of soluble protein levels in muscle.

We next performed histological analysis of the skeletal muscle using H&E and Masson's Trichrome staining. In line with the reduction in muscle mass, we observed severe degeneration and fibrosis of gastrocnemius muscle in Dex-treated control mice, while msSIRT6-KO mice did not show degenerative or fibrotic changes at basal as well Dex-treated conditions (Fig. 4d, e). Besides, the examination of CSA of tibialis anterior muscle myofibers using wheat germ agglutinin (WGA) staining revealed that the msSIRT6-KO mice were protected against Dex-induced reduction in CSA of myofibers (Fig. 4d, f, and Supplementary Fig. 13A). Indeed, we observed increased CSA of myofibers in msSIRT6-KO mice even under basal conditions (Fig. 4d, f, and Supplementary Fig. 13A). In line with this, analysis of muscle fiber size distribution indicated a shift towards smaller myofibers upon Dex-treatment in control mice, while msSIRT6-KO mice showed myofiber enlargement at basal conditions, that were also resistant to Dex-induced shift towards smaller myofibers (Fig. 4d, g, and Supplementary Fig. 13A). We further analyzed the changes in muscle fiber type

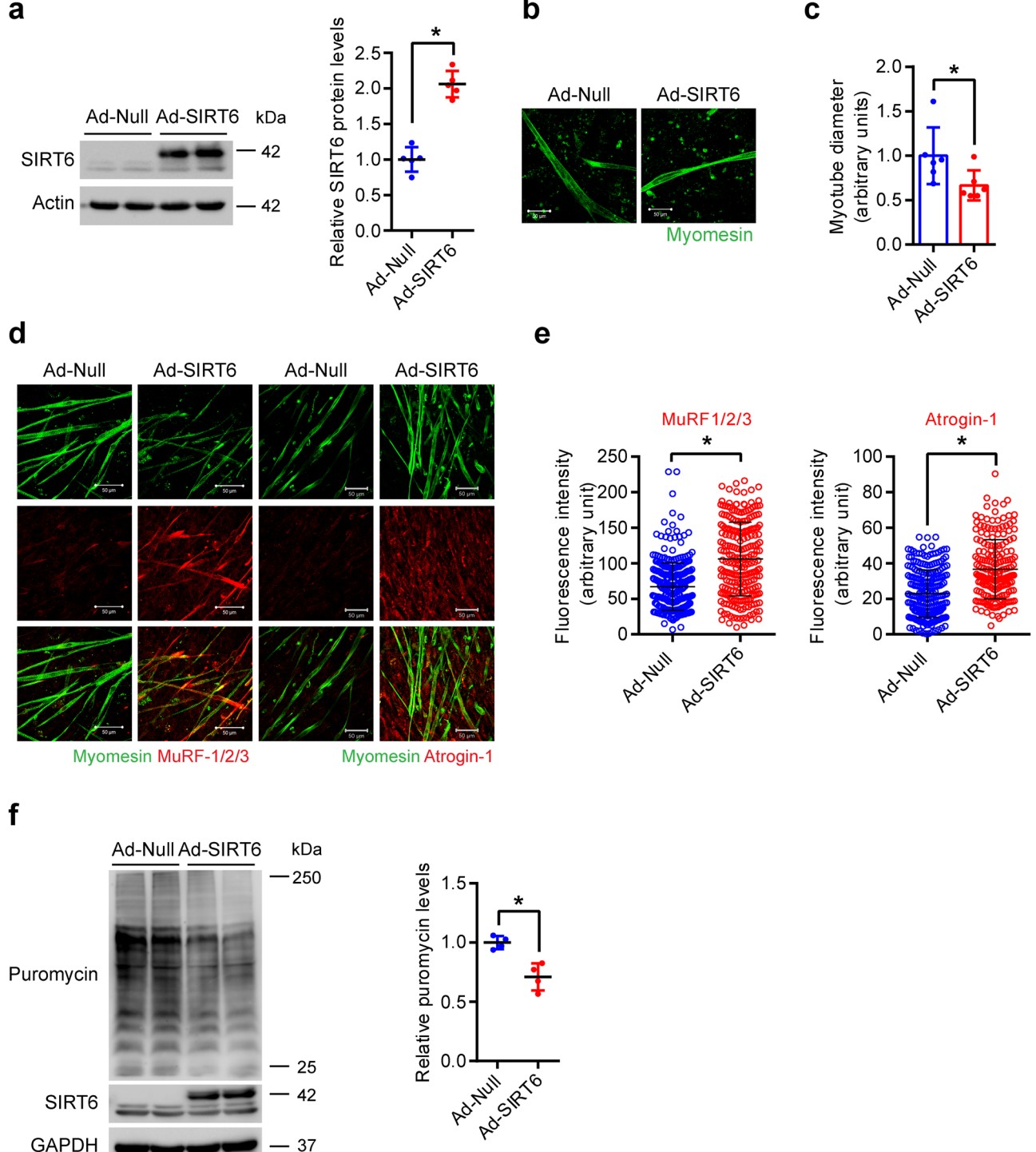

**Fig. 2 | SIRT6 overexpression reduces myotube size. a** Western blot (left) and relative quantification (right) to confirm overexpression of SIRT6 in adenovirus-infected myotubes. $n = 5$. **b** Representative confocal images depicting reduction in myotube size upon SIRT6 overexpression in primary myotubes. The myotubes were stained green using an antibody against myomesin. Scale bar = 50 μm. **c** Quantification of fiber diameter for myotube described in **b**. $n = 6$. **d** Representative confocal images of MuRF-1/2/3 or Atrogin-1 levels upon SIRT6 overexpression in primary myotubes. MuRF-1/2/3 or Atrogin-1 were stained in red and myomesin in green. Scale bar = 50 μm. **e** Scatterplot showing MuRF-1/2/3 or Atrogin-1 fluorescence intensity in myotubes described in **d**. MuRF-1/2/3 (Ad-Null $n = 332$, Ad-SIRT6 $n = 242$), Atrogin-1 (Ad-Null $n = 235$, Ad-SIRT6 $n = 194$). **f** Western blot (left) and relative quantification (right) of puromycin incorporation in SIRT6 overexpressing primary myotubes. $n = 4$. Data presented as mean ± s.d., *$p < 0.05$. Two-tailed unpaired Student's $t$-test was used for statistical analysis for **a, c, e, f**. Source data are provided as a Source Data file.

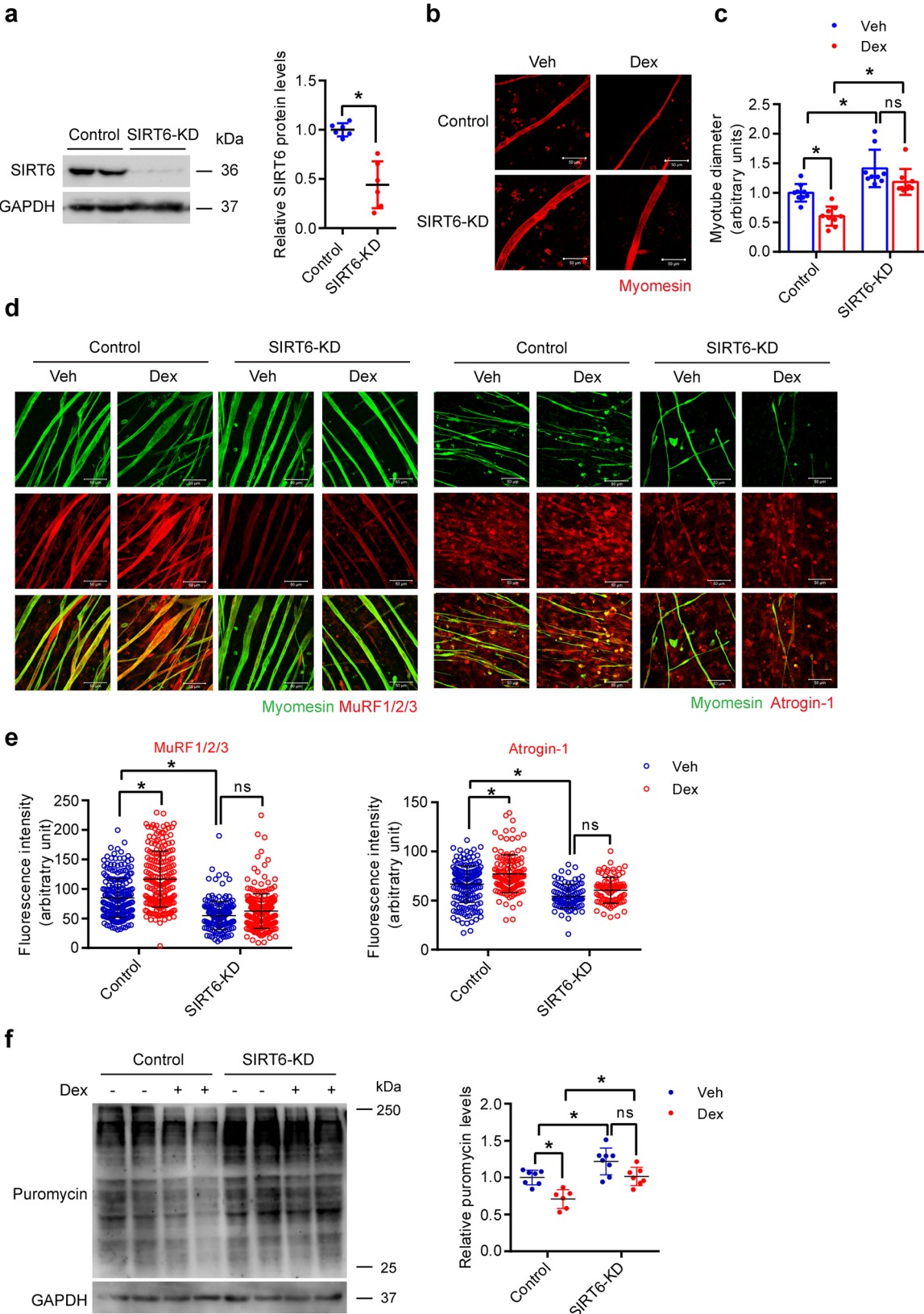

distributions upon Dex treatment in the tibialis anterior of control and msSIRT6-KO mice using the MHC staining. While Dex administration increased the proportion of slow or type MHC I fibers in the tibialis anterior muscle of control mice, the proportion of fast or type MHC IIB fibers was reduced (Supplementary Fig. 13A and B). However, the msSIRT6-KO were resistant to the Dex-induced shift towards slow or type MHC I fibers (Supplementary Fig. 13A and B). Also, we did not find any change in the fiber type proportions in the msSIRT6-KO mice tibialis anterior muscle at the basal level when compared to control mice (Supplementary Fig. 13A and B). Taken together, these observations suggest that msSIRT6-KO mice are protected against Dex-induced muscle wasting and shift in muscle fiber type.

**Fig. 3 | SIRT6 depletion protects against Dex-induced reduction in myotube diameter. a** Western blot (left) and relative quantification (right) to confirm RNAi-mediated depletion of SIRT6 in primary myotube transfected with control siRNA or siRNA targeting SIRT6. $n = 6$. **b** Representative confocal images depicting fiber diameter upon SIRT6 depletion in primary myotubes treated with Dex. Myotubes were stained red using an antibody against myomesin, Scale bar = 50 μm. **c** Quantification of fiber diameter upon SIRT6 depletion in primary myotubes described in **b**. $n = 9$. **d** Representative confocal images of MuRF-1/2/3 or Atrogin-1 levels upon SIRT6 depletion in primary myotubes treated with Dex. MuRF-1/2/3 or Atrogin-1 were stained in red and myomesin in green. Scale bar = 50 μm. **e** MuRF-1/

2/3 or Atrogin-1 fluorescence intensity in myotubes described in **d**. MuRF-1/2/3 (Control-Veh $n = 250$, Control-Dex $n = 204$, SIRT6-KD-Veh $n = 187$, SIRT6-KD-Dex $n = 254$), Atrogin-1 (Control-Veh $n = 173$, Control-Dex $n = 131$, SIRT6-KD-Veh $n = 96$, SIRT6-KD-Dex $n = 92$). **f** Western blot (left) and relative quantification (right) of protein synthesis in Dex or Veh treated SIRT6 depleted myotubes. Control-Veh $n = 7$, Control-Dex $n = 6$, SIRT6-KD-Veh $n = 8$, SIRT6-KD-Dex $n = 7$. Data presented as mean ± s.d., *$p < 0.05$, ns not significant. Two-tailed unpaired Student's $t$-test was used for statistical analysis (**a**), two-way ANOVA with Bonferroni post hoc test (**c**, **e**, **f**). Source data are provided as a Source Data file.

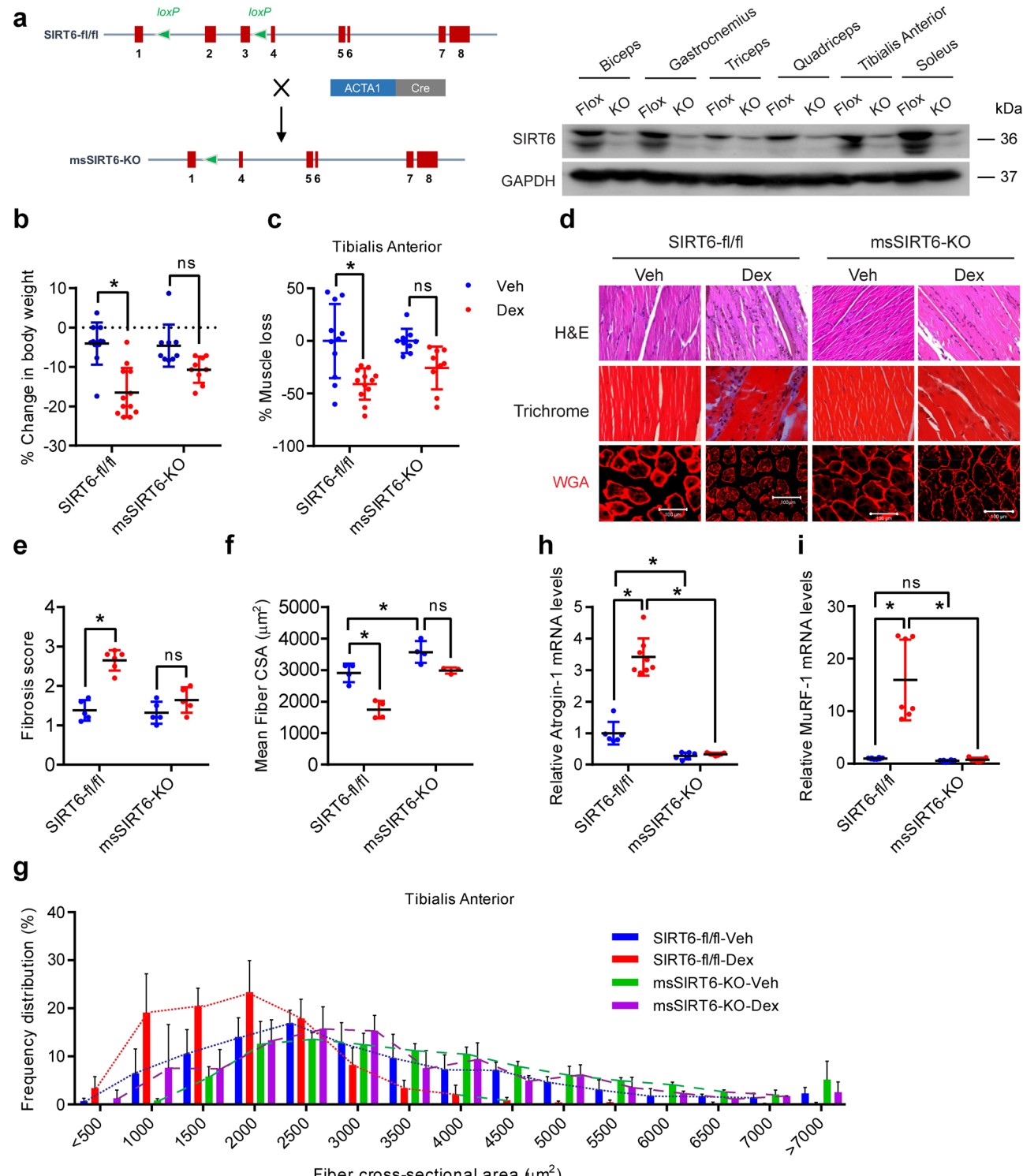

**Fig. 4 | Muscle-specific SIRT6 deficient mice are resistant to Dex-induced muscle atrophy. a** Schematic for generation of muscle-specific SIRT6-KO mice (left) and western blotting (right) showing specific deletion of SIRT6 in biceps, gastrocnemius, triceps, quadriceps, tibialis anterior, and soleus muscles of msSIRT6-KO mice. **b** Change in body weight in SIRT6-fl/fl and msSIRT6-KO mice after Dex or Veh injection. SIRT6-fl/fl-Veh $n = 11$, SIRT6-fl/fl-Dex $n = 12$, msSIRT6-KO-Veh $n = 9$, msSIRT6-KO-Dex $n = 9$. **c** Loss in TA muscle mass in SIRT6-fl/fl and msSIRT6-KO mice after Dex administration. % of muscle loss in SIRT6-fl/fl-Dex or msSIRT6-KO-Dex group is shown relative to Veh injected mice muscle from each genotype. SIRT6-fl/fl-Veh $n = 11$, SIRT6-fl/fl-Dex $n = 12$, msSIRT6-KO-Veh $n = 9$, msSIRT6-KO-Dex $n = 9$. **d** Representative images showing H&E, Masson's Trichrome, and WGA (red) staining in SIRT6-fl/fl and msSIRT6-KO mice muscle after Dex injection. H&E and Masson's Trichrome staining were performed in gastrocnemius muscle. WGA staining was carried out in TA muscle. Scale bar = 100 μm.

**e** Scatterplot showing fibrosis score, scored in a blinded fashion for mice described in **d**. SIRT6-fl/fl-Veh, msSIRT6-KO-Veh, msSIRT6-KO-Dex $n = 5$, SIRT6-fl/fl-Dex $n = 6$. **f** Scatterplot showing mean fiber CSA in mice described in **d**. SIRT6-fl/fl-Veh, SIRT6-fl/fl-Dex, msSIRT6-KO-Veh $n = 4$, msSIRT6-KO-Dex $n = 3$. **g** Frequency distribution of fiber CSA in mice described in **d**. SIRT6-fl/fl-Veh, SIRT6-fl/fl-Dex, msSIRT6-KO-Veh $n = 4$, msSIRT6-KO-Dex $n = 3$. **h** qPCR analysis of Atrogin-1 expression in SIRT6-fl/fl and msSIRT6-KO Dex-treated mice muscle. SIRT6-fl/fl-Veh $n = 6$, SIRT6-fl/fl-Dex $n = 8$, msSIRT6-KO-Veh $n = 6$, msSIRT6-KO-Dex $n = 7$. **i** qPCR analysis of MuRF-1 expression in SIRT6-fl/fl and msSIRT6-KO Dex-treated mice muscle. SIRT6-fl/fl-Veh and msSIRT6-KO-Veh $n = 8$, SIRT6-fl/fl-Dex and msSIRT6-KO-Dex $n = 7$. Data presented as mean ± s.d., *$p < 0.05$, ns not significant. Two-way ANOVA with Bonferroni post hoc test was used for statistical analysis (**b, c, e, f, h, i**). Dex (10 mg/kg/day) was administered to mice. Source data are provided as a Source Data file.

To understand the molecular mechanisms underlying the Dex response, we assessed the expression of atrogenes in muscle tissues of msSIRT6-KO mice treated with Dex. In consonance with the phenotypic and histological observations, the msSIRT6-KO mice were resistant to Dex-mediated induction of atrogenes, while the control mice displayed increased expression of Atrogin-1 and MuRF-1 upon Dex treatment. (Fig. 4h, i). We also observed a significant reduction in the mRNA expression of Atrogin-1, but not of MuRF-1 in the muscles of msSIRT6-KO mice even under basal conditions (Fig. 4h, i). Collectively, these findings reveal that SIRT6 deficiency protects the mice against glucocorticoid-induced atrophic changes in muscle, which in turn is associated with reduced expression of atrogenes.

In order to validate these results in an orthogonal manner, we also analyzed an independent conditional SIRT6 deleted mouse line, generated using a different floxed allele[30] crossed with a Myogenin-Cre transgenic line (Fig. 5a, b). Consistent with our observations in the previous msSIRT6KO line, we did not observe any differences in weight, body mass composition, or blood glucose levels, even after a Glucose Tolerance Test when comparing mice in which SIRT6 was deleted (FF myoCre) and mice heterozygous for the floxed allele (F + myoCre) (Fig. 5c–e). Notably, we observed enlargement of myofibers at basal conditions in these animals, further validating a role of SIRT6 in modulating fiber size (Fig. 5f, g). Furthermore, our preliminary analysis using COX staining suggested enlarged fast fiber types in both ACTA cre and Myogenin cre-generated msSIRT6-KO mice line (Supplementary Fig. 14A). Therefore, we speculate these changes might be fiber type-specific. To address whether this change affects muscle function in vivo, we performed a treadmill exhaustion test in 19 months old mice. We analyzed four FF myoCre and four F + myoCre. We recorded the time and distance that the mice run on the treadmill, and energy expenditure (EE) during the run (Fig. 6). Although we observed a trend for the FF myoCre mice to run less than their F + myoCre littermates (Fig. 6b), when we analyzed the mice altogether, the difference was not statistically significant (Fig. 6c), and energy expenditure was not different between genotypes (Fig. 6d). These data suggest that deletion of SIRT6 is not sufficient to elicit any functional defect in muscle.

## SIRT6 deficiency impairs FoxO transcription factor activity

We found that SIRT6 overexpression is sufficient to increase the expression of Atrogin-1 and MuRF-1/2/3 even under basal conditions (Fig. 2d, e), while SIRT6 deficiency abrogates the increased expression of Atrogin-1 and MuRF-1 in the muscle tissues of Dex-treated mice (Fig. 4h, i). Next, we focused on dissecting the role of SIRT6 in the maintenance of muscle homeostasis in response to Dex treatment. Since SIRT6 is a chromatin-associated deacetylase and is known to control the expression of several target genes by binding to their promoters[16], we first tested whether SIRT6 directly binds to the promoter and controls the expression of Atrogin-1 and MuRF-1. However, our chromatin immunoprecipitation (ChIP) experiments suggest that

SIRT6 is not significantly enriched at the promoters of Atrogin-1 or MuRF-1 (Fig. 7a), indicating that SIRT6 might indirectly regulate the expression of atrogenes.

Previous studies suggest that Atrogin-1 and MuRF-1 are transcriptionally regulated by FoxO transcription factors[7,31]. We hypothesized that SIRT6 deficiency might control atrogene expression by regulating the FoxO transcription factor, which is a key player in muscle atrophy. We therefore tested whether SIRT6 regulates FoxO transcriptional activity using a FoxO responsive luciferase reporter. Interestingly, our results suggest that the transcriptional activity of FoxO is significantly reduced in SIRT6-depleted myotubes (Fig. 7b). Moreover, our ChIP experiments suggest that the binding of FoxO3 to Atrogin-1 and MuRF-1 gene promoters were significantly reduced in msSIRT6-KO mice muscle tissues (Fig. 7c). These findings suggest that SIRT6 might regulate the transcriptional activity of FoxO and its occupancy at the promoters of Atrogin-1 and MuRF-1.

Since the activity of FoxO is controlled by its nuclear localization, we tested whether the localization of FoxO transcription factors could be affected by SIRT6. Reduction of FoxO1 and FoxO3 fluorescence intensity were observed in FoxO1 and FoxO3 depleted myotubes respectively, confirming the specificity of FoxO1 and FoxO3 antibodies (Supplementary Fig. 15A and B). Interestingly, we found that the nuclear localization of FoxO1, as well as FoxO3, were increased in SIRT6 overexpressing myotubes (Fig. 7d, e). Moreover, we found that the total protein levels of both FoxO1 and FoxO3 were also significantly increased in SIRT6 overexpressing myotubes (Fig. 7d, f, g). On the other hand, SIRT6 depletion reduced the expression of FoxO1 and FoxO3 under basal as well as Dex-treated conditions in myotubes (Fig. 8a–c). Similarly, we also found that the mRNA levels of FoxO1 and FoxO3 were decreased in the gastrocnemius muscles of msSIRT6-KO mice both under basal and Dex-treated conditions (Fig. 8d, e). Overall, these data suggest that SIRT6 could regulate the expression of atrogenes by controlling both the expression level and the nuclear localization of FoxO1 and FoxO3 isoforms.

To test whether reduced FoxO activity is the major reason for resistance against reduction in myotube size under SIRT6 depleted conditions, we overexpressed a constitutively active FoxO in SIRT6 depleted myotubes and assessed the myotube diameter. Notably, we found that expressing a constitutively active FoxO was sufficient to induce a reduction in the diameter of SIRT6 depleted myotube (Fig. 8f, g). Collectively, these findings suggest that SIRT6 deficiency protects the myotubes from Dex-induced reduction in size majorly by attenuating the transcriptional activity of FoxO transcription factors.

## SIRT6 deficiency hyperactivates IGF/PI3K/AKT signaling in skeletal muscle

A key regulator of the nuclear localization of FoxO and its stability is the IGF/AKT signaling. Increased AKT activation results in enhanced phosphorylation of FoxO which prevents its nuclear retention and leads to subsequent degradation[32]. Notably, we have previously shown

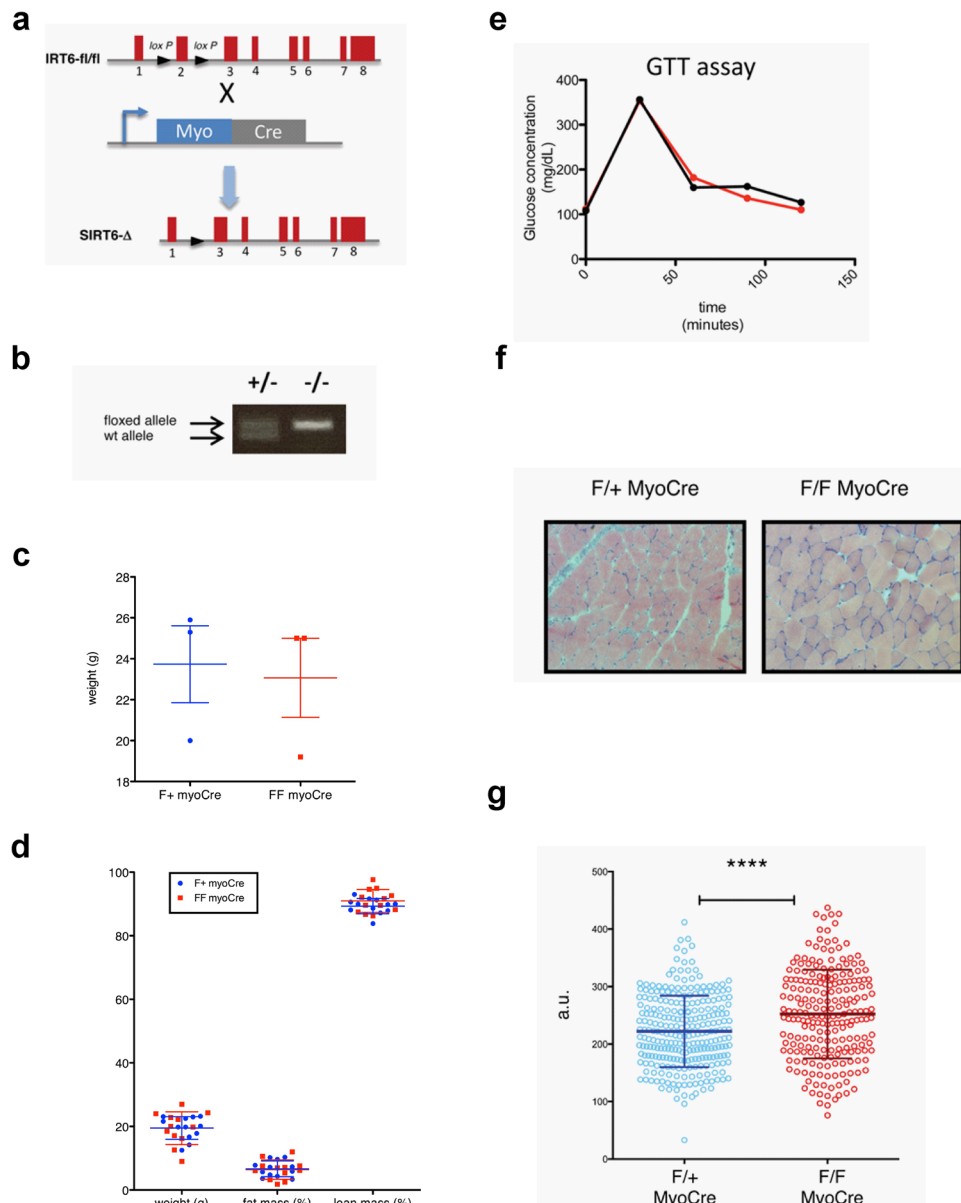

**Fig. 5 | Phenotypic characterization of muscle-specific SIRT6 deficient mice line generated using myogenin cre. a** Diagram showing the deletion of Sirt6 in skeletal muscle. C57BL/6 mice containing floxed alleles of Sirt6[30] were crossed with transgenic mice carrying cre under the control of the muscle-specific myogenin promoter. Myogenin-Cre triggered the recombination of the floxed allele resulting in the deletion of sirt6 specifically in muscle. **b** PCR of genomic DNA extracted from the tail of myogenin cre mice that were heterozygous (F/+ MyoCre) or homozygous (F/F MyoCre) for the floxed sirt6. **c** Body weight of F/+ MyoCre and F/F MyoCre mice. $n = 3$. **d** Body composition of F/+ MyoCre and F/F MyoCre mice. $n = 12$.

**e** Glucose Tolerance Test (GTT) were analyzed in 4-months old SIRT6 F/+ MyoCre control and SIRT6 F/F MyoCre mice, as indicated. **f** Soleus and gastrocnemius muscle from 5 months old F/+ MyoCre and F/F MyoCre mice were collected in OCT and cross-sections were stained with H&E to determine the fiber diameter. 2 mice per genotype. **g** The maximum diameter of over 200 fibers obtained from 2 mice per genotype was determined in ImageJ and the two data sets were compared by unpaired two tailed *t*-test. The *p* value was <0.0001 (****). Data presented as mean ± s.d., Two-tailed unpaired Student's *t*-test was used for statistical analysis (**c, d, g**). Source data are provided as a Source Data file.

that SIRT6 deficiency leads to increased AKT signaling in the heart[20]. Since FoxO levels and transcriptional activity were reduced in SIRT6 depleted myotubes (Figs. 7b, 8a–c), we tested the status of AKT signaling in msSIRT6-KO mice muscles by western blotting. We found that the phosphorylation of AKT and the downstream target mTOR were increased not only at the basal level but also in Dex-treated conditions in msSIRT6-KO muscle when compared to corresponding controls (Fig. 9a and Supplementary Fig. 16A, B, H). In addition, the total protein levels of AKT and mTOR were also increased under SIRT6 deficient conditions (Fig. 9a and Supplementary Fig. 16C, I), which is consistent with our previous reports[20,33]. In line with the increased activation of AKT, we found increased inhibitory phosphorylation of

FoxO1 and FoxO3 and reduced total protein levels of both FoxO1 and FoxO3 in the muscles of Veh as well as Dex-treated msSIRT6-KO mice (Fig. 9a and Supplementary Fig. 16D–G). These results are consistent with the reduced FoxO levels and activity observed in the SIRT6 depleted myotubes (Figs. 7b, 8a–c). We next analyzed the protein synthesis rates by SUnSET analysis in the muscle of msSIRT6-KO mice, with or without 15 days of Dex administration as protein synthesis is a major determinant of muscle mass and phenotype. Interestingly, we found an increased rate of protein synthesis in msSIRT6-KO mice in basal as well as Dex-treated conditions (Fig. 9a and Supplementary Fig. 16J–L). In line with our previous results (Supplementary Fig. 7A), Dex administration for 15 days reduced SIRT6 levels in mice

**a**

| Time [s] | Start Speed [m/s] | End Speed [m/s] |
|---|---|---|
| 30 | 0 | 0.25 |
| 300 | 0.25 | 0.25 |
| 30 | 0.25 | 0.35 |
| 300 | 0.35 | 0.35 |
| 30 | 0.35 | 0.45 |
| 300 | 0.45 | 0.45 |
| 30 | 0.45 | 0.55 |
| 300 | 0.55 | 0.55 |

**c**

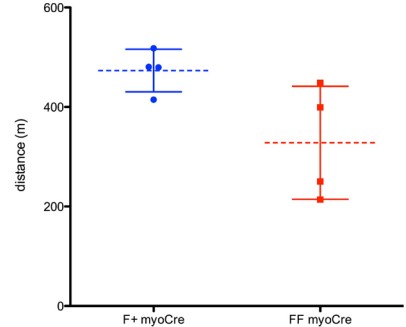

**b**

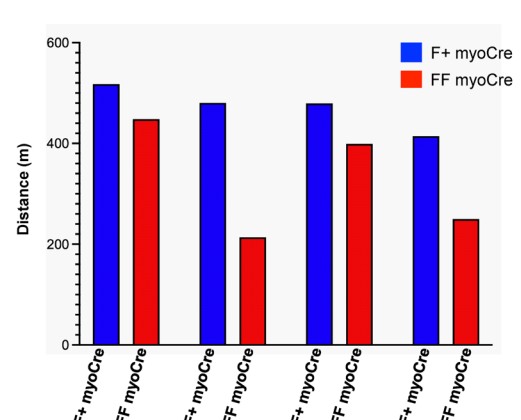

**d**

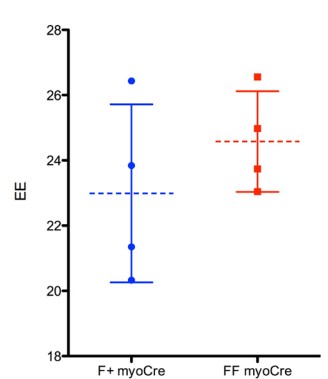

**Fig. 6 | SIRT6 deletion in muscle does not significantly affect muscle performance in vivo.** The experiment was performed on 4 mice per genotype. **a** Table summarizing the setting for the muscle performance experiment. In brief, mice ran up to 300 s at each speed. 30 s were allowed to increase the speed between the 300 s intervals. **b** The histogram represents the distance each pair of mice (F + myoCre and same gender FF myoCre littermate) ran. **c** Mean distance +/− SD that F + myoCre and FF myoCre mice ran. **d** Mean energy expenditure during the treadmill run +/− SD. Data presented as mean ± s.d., *$p < 0.05$, ns not significant. Two-tailed paired Student's $t$-test was used for statistical analysis (**c**, **d**). Source data are provided as a Source Data file.

gastrocnemius muscle (Fig. 9a and Supplementary Fig. 16M). AMPK is another key regulator of muscle growth and development that controls both muscle atrophy and hypertrophy signaling[34]. AMPK promotes muscle protein degradation by increasing the transcriptional activity of FoxO, while inhibiting protein synthesis by negatively regulating mTOR signaling[35]. In line with reduced levels of FoxO1 and FoxO3, we found reduced activating phosphorylation of AMPK in msSIRT6-KO mice muscle at basal level (Fig. 9a and Supplementary Fig. 16N and O). Moreover, msSIRT6-KO mice were resistant to Dex-induced increase in AMPK phosphorylation (Fig. 9a and Supplementary Fig. 16N and O). We further analyzed the change in AMPK activity by testing ACC phosphorylation, an AMPK downstream target in Dex-treated msSIRT6-KO. While Dex-treated control mice showed increased ACC phosphorylation, msSIRT6-KO mice were resistant against Dex-induced increase in ACC phosphorylation. Furthermore, the level of ACC phosphorylation was comparable to controls at basal level which further emphasises the role of AMPK in SIRT6 mediated regulation of muscle mass (Supplementary Fig. 17A).

Next, we tested whether the reduced AMPK activity in msSIRT6-KO mice muscle is the major player for protection against Dex-induced muscle atrophy or is simply the secondary effect of prolonged absence of SIRT6 by performing in vitro experiments using compound C (AMPK inhibitor). We tested whether AMPK inhibitor compound C could rescue reduction in SIRT6 overexpressing primary myotube diameter. Interestingly, we found that AMPK inhibitor compound C could not rescue SIRT6 overexpression-induced reduction in myotube diameter (Supplementary Fig. 17B–D). In addition, inhibition of AMPK

spontaneously led to a reduction in myotube diameter (Supplementary Fig. 17B–D). These results suggest that long-term deficiency of SIRT6 might be responsible for changes in AMPK signaling in msSIRT6-KO mice muscle. Therefore, we speculate that AMPK signaling might not be directly involved in mediating the regulatory action of SIRT6 on muscle.

To better understand the mechanism of activation of AKT signaling, we tested the expression levels of various positive and negative regulators of the pathway. Notably, we found that the mRNA levels of IGF2, a key activator of the IGF receptor that controls AKT activation[36], was significantly increased in the muscles of msSIRT6-KO mice (Fig. 9b). In line with this, we also observed increased protein levels of IGF2 and enhanced phosphorylation of the IGF receptor (Fig. 9c and Supplementary Fig. 18A). These findings suggest that SIRT6 deficiency hyperactivates IGF/AKT signaling cascade via increased IGF2 levels to regulate muscle homeostasis.

To test whether SIRT6 regulates AKT signaling in a deacetylation-dependent manner, we ectopically expressed wild-type or catalytically inactive SIRT6 and assessed the AKT signaling. We found that wild-type SIRT6, but not catalytically inactive SIRT6 suppresses the AKT signaling, suggesting the requirement of enzymatic activity (Fig. 9d and Supplementary Fig. 18B–E). Finally, we tested whether inhibition of AKT signaling is sufficient to abrogate the protection of SIRT6-depleted myotubes against Dex-induced reduction in size. Notably, we found that treatment of SIRT6 depleted myotubes with the AKT inhibitor AKTi-1/2 or the PI3K inhibitor LY294002 indeed renders them susceptible to reduction in diameter

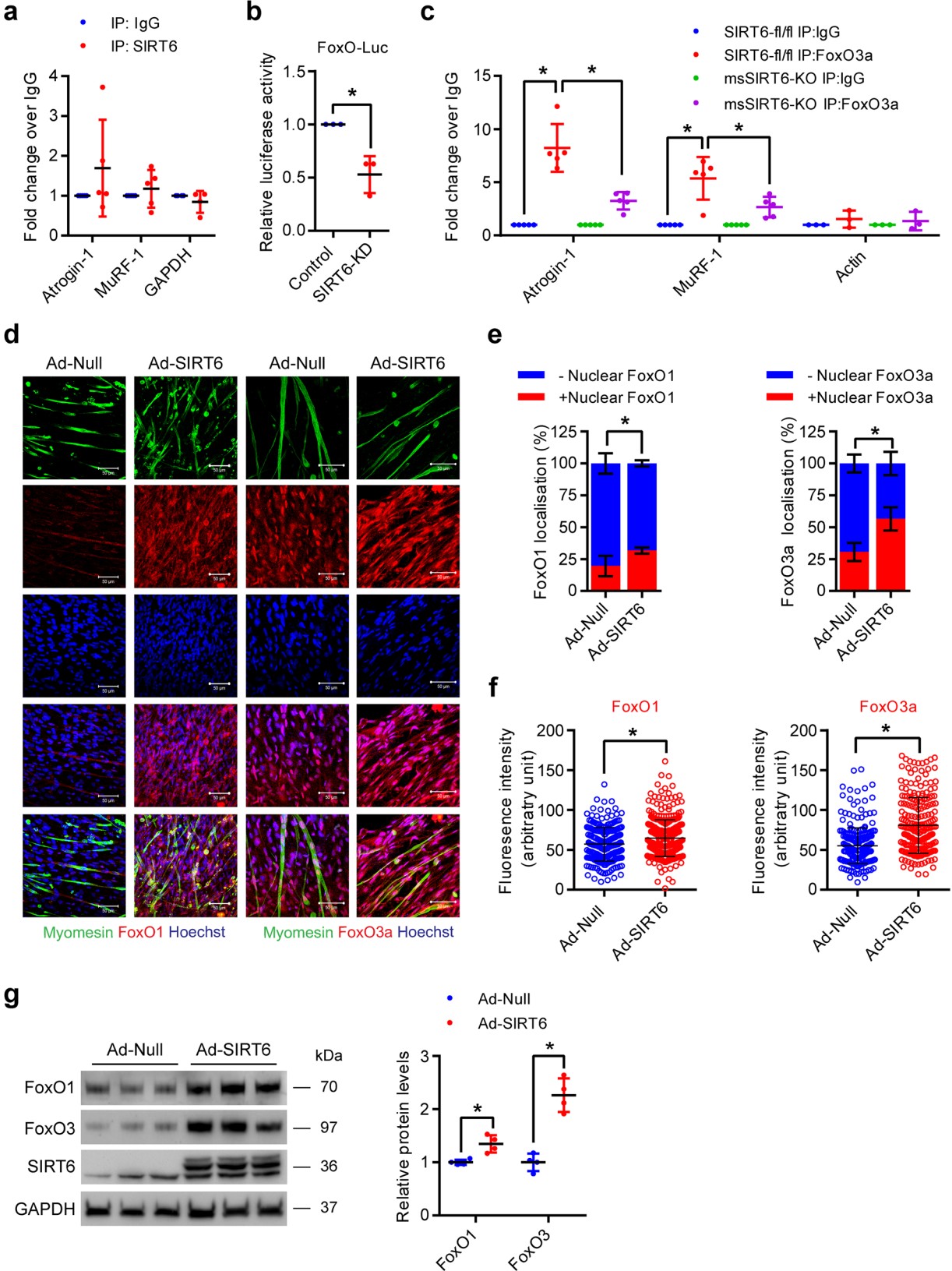

upon Dex treatment (Fig. 9e, f and Supplementary Fig. 18F–I). Though PI3K inhibitor LY294002 shows off-target effects[37], AKT inhibitor AKTi-1/2 is a highly selective non-competitive inhibitor of AKT[37]. Therefore, comparing the results of both the inhibitors together suggest that SIRT6 impacts myotube physiology via modulation of the AKT activity.

## SIRT6 represses c-Jun transcription factor to control muscle atrophy

Our results indicate that IGF2 is transcriptionally upregulated in msSIRT6-KO mice muscle tissue (Fig. 9b). We, therefore, tested whether SIRT6 binds to the IGF2 promoter. Our ChIP analysis indicated that SIRT6 is significantly enriched at the IGF2 promoter (Fig. 9g).

**Fig. 7 | SIRT6 deficiency reduces FoxO binding to atrogene promoters by regulating its nuclear localization. a** ChIP analysis for SIRT6 binding to Atrogin-1 and MuRF-1 promoters in mice muscle. GAPDH was used as a negative control. Atrogin-1 and MuRF-1 $n = 5$, GAPDH $n = 4$. **b** Relative FoxO luciferase (FHRE-Luc) activity in SIRT6 depleted primary myotubes. $n = 3$. **c** ChIP analysis for FoxO3 binding to Atrogin-1 and MuRF-1 promoters in SIRT6-fl/fl and msSIRT6-KO mice muscle. Actin was used as a negative control. Atrogin-1 and MuRF-1 $n = 5$, Actin $n = 3$. **d** Representative confocal images showing subcellular localization and levels of FoxO1 and FoxO3 in SIRT6 overexpressing primary myotubes. FoxO1 or FoxO3 and myomesin were stained red and green respectively. Nuclei are stained blue with Hoechst 33342. Scale bar = 50 μm. **e** Graph showing FoxO1 and FoxO3 nuclear localization in myotubes described in **d**. $n = 4$. **f** Scatterplot showing FoxO1 and FoxO3 fluorescence intensity in myotubes described in **d**. FoxO1 (Ad-Null $n = 241$, Ad-SIRT6 $n = 381$), FoxO3 (Ad-Null $n = 276$, Ad-SIRT6 $n = 227$). **g** Western blot (left) and relative quantification (right) of FoxO1 and FoxO3 protein levels in SIRT6 overexpressing primary myotubes. $n = 4$. Data presented as mean ± s.d., *$p < 0.05$. Two-tailed unpaired Student's $t$-test was used for statistical analysis **a–c**, **e–g**. Source data are provided as a Source Data file.

Notably, it has been shown that IGF2 is regulated by the AP-1 complex in skeletal muscle[38]. Also, we have previously found that the c-Jun transcription factor which is part of the AP-1 complex is hyperactive in SIRT6 deficient cardiomyocytes and is responsible for higher expression levels of IGF2 in the heart[20]. Based on these observations, we, therefore, tested the interaction of SIRT6 with c-Jun in myotubes. Our immunoprecipitation experiments revealed that SIRT6 indeed binds to c-Jun in myotubes at endogenous levels (Supplementary Fig 18J). Moreover, we found that SIRT6 depletion enhanced the c-Jun occupancy at the IGF2 promoter, indicating a key role for SIRT6 in regulating IGF2 expression (Fig. 9h). Since SIRT6 is known to regulate the occupancy of transcription factors at the promoter regions by modifying the histone acetylation status[39], we next tested the acetylation level of SIRT6 target residue histone 3 lysine 9 at the IGF2 promoter. Our results indicate that depletion of SIRT6 significantly increases the acetylation of H3K9 residues around the IGF2 promoter (Fig. 9i). Taken together, these data suggest that SIRT6 regulates the expression of IGF2 in muscle by modulating the acetylation status of H3K9 and controlling the occupancy of c-Jun.

Since we found that c-Jun is involved in the activation of IGF/AKT signaling in SIRT6 deficient muscle, we tested whether the inhibition of endogenous c-Jun could cause a reduction in the size of SIRT6 depleted myotubes[40]. Confocal image analysis shows that inhibition of endogenous c-Jun indeed led to a decrease in the diameter of SIRT6 depleted primary myotubes (Fig. 9j, k), indicating the key role of c-Jun in regulating skeletal muscle phenotype under SIRT6 deficient conditions.

### Pharmacological inhibition of SIRT6 protects mice against Dex-induced muscle atrophy

To explore the clinical relevance of these findings, we tested whether pharmacological inhibition of SIRT6 could be used to protect mice against Dex-induced muscle atrophy. To test this, we administered vehicle as well as Dex-treated mice with the SIRT6 inhibitor 2,4-dioxo-N-(4-(pyridin-3-yloxy)phenyl)−1,2,3,4-tetrahydroquinazoline-6-sulfo-namide which has been described previously[41,42]. The compound inhibits SIRT6 activity by binding to the NAM and substrate binding pockets[41]. In line with this, we found that administration of SIRT6 inhibitor increased the acetylation levels of histone 3 at lysine 9 residue in the muscle, confirming the inhibitory effect of the compound on SIRT6 activity (Fig. 10a). However, the acetylation levels of SOD2 or H3K18, targets of SIRT3[43,44] and SIRT7[44,45] respectively, were unaltered in SIRT6 inhibitor-treated mice muscle (Fig. 10a). In addition, we have previously shown that the acetylation levels of SIRT1 and SIRT2 targets, p53 and α-tubulin respectively also do not change in SIRT6 inhibitor-treated mice muscle[42]. These data suggest the high specificity of the SIRT6 inhibitor in muscle. Moreover, we did not observe any significant change in the mRNA or protein levels of the various Sirtuin isoforms except for the reduction in SIRT2 mRNA levels in SIRT6 inhibitor-treated mice muscle (Supplementary Fig. 19A and B), the reason for which is unclear. We did not investigate this further as the protein levels of SIRT2 was not changed.

Consistent with our findings in msSIRT6-KO mice, we found that inhibition of SIRT6 prevented the decrease in bodyweight upon Dex administration (Fig. 10b and Supplementary Fig. 19C). Similarly, we found that SIRT6 inhibition abrogated the decrease in the muscle weight of tibialis anterior and gastrocnemius muscles upon Dex administration (Fig. 10c and Supplementary Fig. 19D–F). However, no significant differences were observed in other muscle types (quadriceps, biceps, and triceps) due to SIRT6 inhibitor or Dex treatment (Supplementary Fig. 19G–L). We also found that treatment with a SIRT6 inhibitor increased the myofiber CSA of the tibialis anterior muscle at the basal level and protected against Dex-induced reduction in myofiber CSA (Fig. 10d, e and Supplementary Fig. 20A). Moreover, SIRT6 inhibitor-treated mice muscles showed a shift towards larger myofiber at the basal level (Fig. 10d, f, and Supplementary Fig. 20A), and similar to msSIRT6-KO mice, SIRT6 inhibitor-treated mice muscles were protected from Dex-induced shift towards slow fiber types (Supplementary Fig. 20A and B). Collectively, these results suggest that similar to muscle-specific deletion of SIRT6, inhibition of SIRT6 using pharmacological inhibitor also protects muscle against Dex-induced atrophy and shift in fiber types.

Since Dex-induced muscle loss in control mice, while SIRT6 inhibitor-treated mice were protected against Dex-induced muscle atrophy, we next tested the effect of Dex on the activity of SIRT6 in a vehicle or SIRT6-inhibitor-treated mice by evaluating the acetylation status of H3K9. Our result showed that Dex indeed reduced the acetylation level of H3K9 at the basal level (Fig. 10g and Supplementary Fig. 21I). However, Dex also decreased the acetylation level of H3K9 in inhibitor-treated mice muscle (Fig. 10g and Supplementary Fig. 21I), suggesting Dex induces muscle loss by modulating SIRT6 activity. Furthermore, these results indicate a complex regulation of the acetylation status of H3K9 in response to Dex treatment. Notably, Dex has been shown to reduce the acetylation level of H3K9 by controlling its upstream regulators, p300 and HDAC3[46,47]. Therefore, future studies are needed to understand the complex interplay between histone acetylases and deacetylases in Dex-treated muscle.

We next tested the effect of the SIRT6 inhibitor on the rate of muscle protein synthesis. Notably, SUnSET analysis showed that SIRT6 inhibition increases the muscle protein synthesis in basal as well as Dex-treated conditions as observed in msSIRT6-KO mice (Fig. 10g, Supplementary Fig. 21A–B). Similar to msSIRT6-KO mice, SIRT6-inhibitor-treated mice show increased activation of AKT signaling, which results in the inactivation of FoxO and its degradation (Fig. 10g, Supplementary Fig. 21C–G). We also tested AMPK activity in Dex or Veh administered SIRT6 inhibitor-treated mice. Our results suggest that phosphorylation of ACC in Dex-administered SIRT6 inhibitor-treated mice was comparable to control Dex-treated mice (Supplementary Fig. 21H). However, phosphorylation of ACC was low in SIRT6 inhibitor-treated mice at the basal level (Supplementary Fig. 21H). These results suggest that AMPK signaling might not be a crucial player in protection from Dex-induced muscle atrophy in SIRT6 inhibitor-treated mice. Furthermore, a shorter duration of SIRT6 inhibitor-treated mice might be protected from the secondary effect of SIRT6 deficiency in muscle under stressed conditions. Collectively, these results suggest that SIRT6 inhibition protects the muscles from Dex-induced atrophy by enhancing protein synthesis and inhibiting FoxO1 through activation of IGF/AKT signaling.

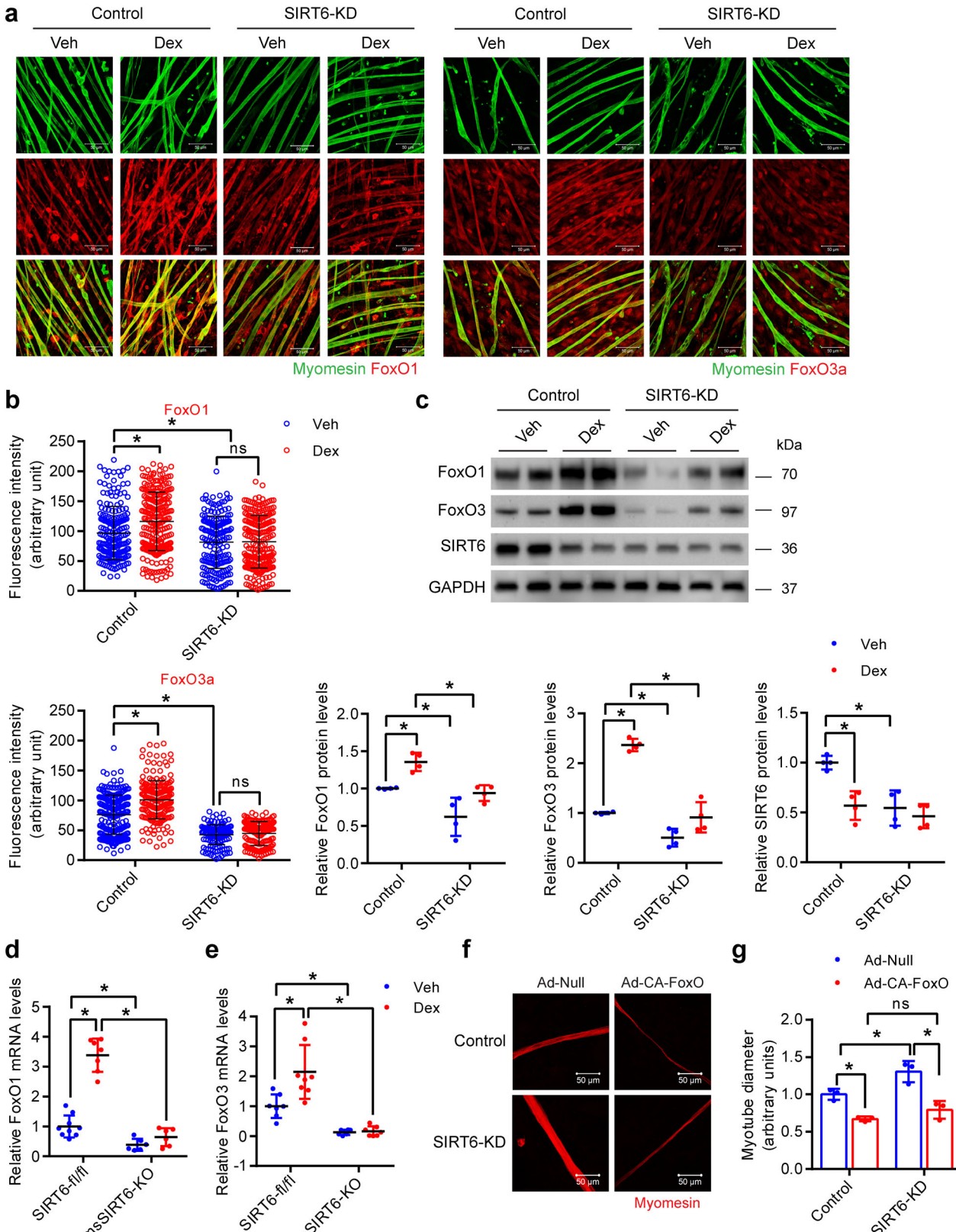

## Discussion

In the present work, we uncovered a crucial role for SIRT6 in regulating skeletal muscle atrophy using primary myotubes and two independent lines of muscle-specific SIRT6-KO mice. Importantly, we find that SIRT6 inhibition preserves muscle mass and protects against glucocorticoid-induced atrophy of skeletal muscles by directly controlling the activity of PI3K/AKT signaling which is involved in regulating both muscle protein synthesis and degradation.

Different Sirtuins have been shown to exert varied effects in the muscle in response to different physiological and pathological cues. It has been previously shown that SIRT1 promotes muscle growth and protects against fasting and denervation-induced muscle atrophy by

**Fig. 8 | Constitutively active FoxO reduces SIRT6 depleted primary myotube diameter. a** Representative confocal image depicting FoxO1 (left) and FoxO3 (right) levels in SIRT6 depleted primary myotubes treated with Dex. FoxO1 or FoxO3 and myomesin were stained red and green respectively. Scale bar = 50 μm. **b** Quantification of FoxO1 (top) and FoxO3 (bottom) fluorescence intensity in myotubes described in **a**. FoxO1 (Control-Veh $n = 201$, Control-Dex $n = 242$, SIRT6-KD-Veh $n = 182$, SIRT6-KD-Dex $n = 228$), FoxO3 (Control-Veh $n = 220$, Control-Dex $n = 219$, SIRT6-KD-Veh $n = 146$, SIRT6-KD Dex $n = 152$). **c** Western blot (top) and relative quantification (bottom) of FoxO1, FoxO3 and SIRT6 protein levels in Dex-treated SIRT6 depleted myotubes. FoxO1, FoxO3 and SIRT6 $n = 4$. **d** qPCR analysis of FoxO1 levels in SIRT6-fl/fl and msSIRT6-KO mice muscle after Dex treatment.

SIRT6-fl/fl-Veh $n = 9$, SIRT6-fl/fl-Dex $n = 7$, msSIRT6-KO-Veh and msSIRT6-KO-Dex $n = 6$. **e** qPCR analysis of FoxO3 levels in SIRT6-fl/fl and msSIRT6-KO mice muscle after Dex adminstration. SIRT6-fl/fl-Veh $n = 7$, SIRT6-fl/fl-Dex $n = 8$, msSIRT6-KO-Veh $n = 6$, msSIRT6-KO-Dex $n = 7$. **f** Representative confocal images depicting diameter of SIRT6 depleted primary myotubes infected with either constitutively active FoxO (Ad-CA-FoxO) or null adenovirus. The myotubes were stained red using antibody against myomesin. Scale bar = 50 μm. **g** Quantification for fiber diameter in myotube described in **f**. $n = 3$. Data presented as mean ± s.d., *$p < 0.05$, ns not significant. Two-way ANOVA with Bonferroni post hoc test was used for statistical analysis (**b–e**, **g**). Dex (10 mg/kg/day) was administered (**d**, **e**). Source data are provided as a Source Data file.

suppressing FoxO1 and FoxO3-mediated atrogene expression[48]. Similarly, SIRT2 protects against glucocorticoid-induced reduction in myotube diameter by inhibiting autophagy[49]. With respect to SIRT6, a previous study observed that SIRT6 deficiency leads to muscle atrophy[50]. However, the authors derived this conclusion based on studies in a whole body SIRT6 knockout mice model. These mice exhibit severe metabolic defects including severe hypoglycemia and display accelerated aging like phenotype[17]. It is therefore difficult to establish whether the observed muscle phenotype is a direct outcome of SIRT6 deficiency, or it is due to secondary effects resulting from systemic complications such as severe hypoglycemia. To overcome this limitation, we developed two independent lines of muscle-specific SIRT6-KO mice, which would be highly relevant to studying the role of SIRT6 in skeletal muscle. In contrast to the observations in whole body SIRT6 knockout mice[50], we find that muscle-specific deletion of SIRT6 has no evident effect on muscle mass and protects mice against glucocorticoid-induced muscle wasting.

Studies indicate that AKT1 and AKT2 isoforms are essential for maintaining muscle mass and the AKT2 isoform accounts for 90% of muscle AKT[51,52]. Mice lacking AKT1 or AKT2 show reduced muscle mass, contraction force, and grip strength[52]. In line with this, AKT1/AKT2-double-KO mice show severe growth deficiency and die soon after birth[53]. Further, AKT1/AKT2-double KO mice develop skeletal muscle atrophy[53]. In a similar line, skeletal muscle-specific AKT2 deficient mice show muscle weight comparable to controls due to a compensatory increase in AKT1 level[51]. Moreover, skeletal muscle-specific deletion of AKT1 and AKT2 results in a significant reduction in muscle mass and muscle CSA[51]. Another report suggests that AKT2 plays an important role in skeletal muscle development and glucose metabolism via AMPK signaling[54]. Therefore, AKT2-KO mice develop a reduction in skeletal muscle mass, defects in glucose metabolism, and mitochondria biogenesis due to impaired activation of AMPK signaling[54]. Based on these reports, we speculate that SIRT6 controls the activity of PI3K/AKT signaling while protecting against Dex-induced muscle mass loss through the combined action of AKT1/AKT2.

SIRT6 deficiency hyperactivates PI3K/AKT signaling in different tissues, although the mechanisms are different. In the liver of SIRT6 transgenic male mice, the mRNA levels of IGF-binding protein 1, which inhibits the activation of IGF1 signaling are high[19]. On the other hand, SIRT6-deficient hearts express higher levels of IGF2, IGF1R, and AKT to hyperactive PI3K/AKT signaling[20]. Consistent with the findings in the heart, our results reveal that muscle-specific SIRT6-deficient muscle show higher levels of IGF2 expression which hyperactivate PI3K/AKT signaling in the skeletal muscle. The increased PI3K/AKT signaling in turn leads to the inactivation of FoxO and thus protects mice against muscle atrophy. Notably, we have shown in our recent study that SIRT6 deficiency leads to transcriptional activation of the mTOR signaling which increases protein synthesis in the muscle[33]. It is possible that SIRT6 deficiency might confer protection against muscle atrophy through the combined action of IGF2-mediated AKT activation and the transcriptional activation of mTOR signaling.

Recent work reported that SIRT6 deficiency in muscle led to impaired insulin signaling[55]. In contrast, our data suggest that muscle-

specific SIRT6 deletion increases AKT signaling. We have also previously shown that SIRT6 deficiency in the heart leads to a similar activation of AKT signaling[20]. In the same line, the treatment of high fat diet-fed mice with a SIRT6 inhibitor significantly increased p-AKT levels in both liver and muscle tissue[42]. While in the same study authors reported that mice deficient in SIRT6 in muscle showed reduced activity of AMPK leading to altered metabolism in mice[55], we did not find any significant difference in metabolic activity in FF myoCre mice at the basal level. The differences in the phenotype could be due to different approaches used to generate the muscle-specific SIRT6 knockout mice. In this work, we generated muscle-specific SIRT6-KO using the Cre recombinase gene driven by the alpha-skeletal actin or myogenin cre, whereas in the other work[55], the authors employed muscle creatinine kinase promoter to delete SIRT6 in the muscle tissue. In contrast to our findings, a recent study has shown that muscle-specific deletion of SIRT6 reprograms the myofibers towards oxidative types and increases muscle performance[56]. The authors have used mice expressing myosin light polypeptide 1 cre (Myl1-Cre) for the generation of muscle-specific knockout mice which might be the reason behind the observed discrepancies in myofiber types.

Similar to muscle-specific SIRT6 deficient mice, we also find that acute inhibition of SIRT6 using a recently developed pharmacological inhibitor protects against glucocorticoid-induced muscle atrophy. Whole-body SIRT6 knockout mice show the absence of SIRT6 mRNA and protein in all cell types. However, the SIRT6 inhibitor specifically inhibits the catalytic activity of SIRT6 without targeting mRNA and protein levels of SIRT6 which will have a restricted impact on the body when compared to whole-body deletion of SIRT6. Emerging roles of SIRT6 suggest its catalytic independent functions in different cell types[33,57]. The idea of using a SIRT6 inhibitor in this study fits well with the deacetylase activity-dependent function of SIRT6 in protection against Dex-induced muscle loss. Therefore, it will not affect those SIRT6 functions which are independent of SIRT6 catalytic activity. Moreover, the effect of the pharmacological inhibitor used in this study is limited to a shorter period in contrast to the generalized long-term impact of whole-body SIRT6 deficiency on the body. In line with this, acute inhibition of SIRT6 using small molecule inhibitors or tissue-specific deletion in mice models has been shown to be beneficial in a context-specific manner. SIRT6 inhibitors have been shown to be beneficial in reducing blood glucose levels and improving insulin tolerance in a diabetic mice model, via enhanced glucose transporter and glycolytic enzyme expression in muscle[42]. Also, the use of the same SIRT6 inhibitor impaired dendritic cell migration, downregulated pathogenic T cell inflammatory responses, and delayed the onset of experimental autoimmune encephalomyelitis, the most common animal model of multiple sclerosis[58]. Moreover, SIRT6 deficiency in rod photoreceptors improves the survival of photoreceptors and preserves vision in a mouse model of retinitis pigmentosa[59]. In addition, SIRT6 deficiency enhanced contextual fear memory without affecting spatial memory in mice[60]. Therefore, SIRT6 inhibition might improve organ function depending on the nature of the stress and disease. However, the usage of SIRT6 inhibitor for protection against Dex-induced muscle atrophy has certain limitations such as off-target

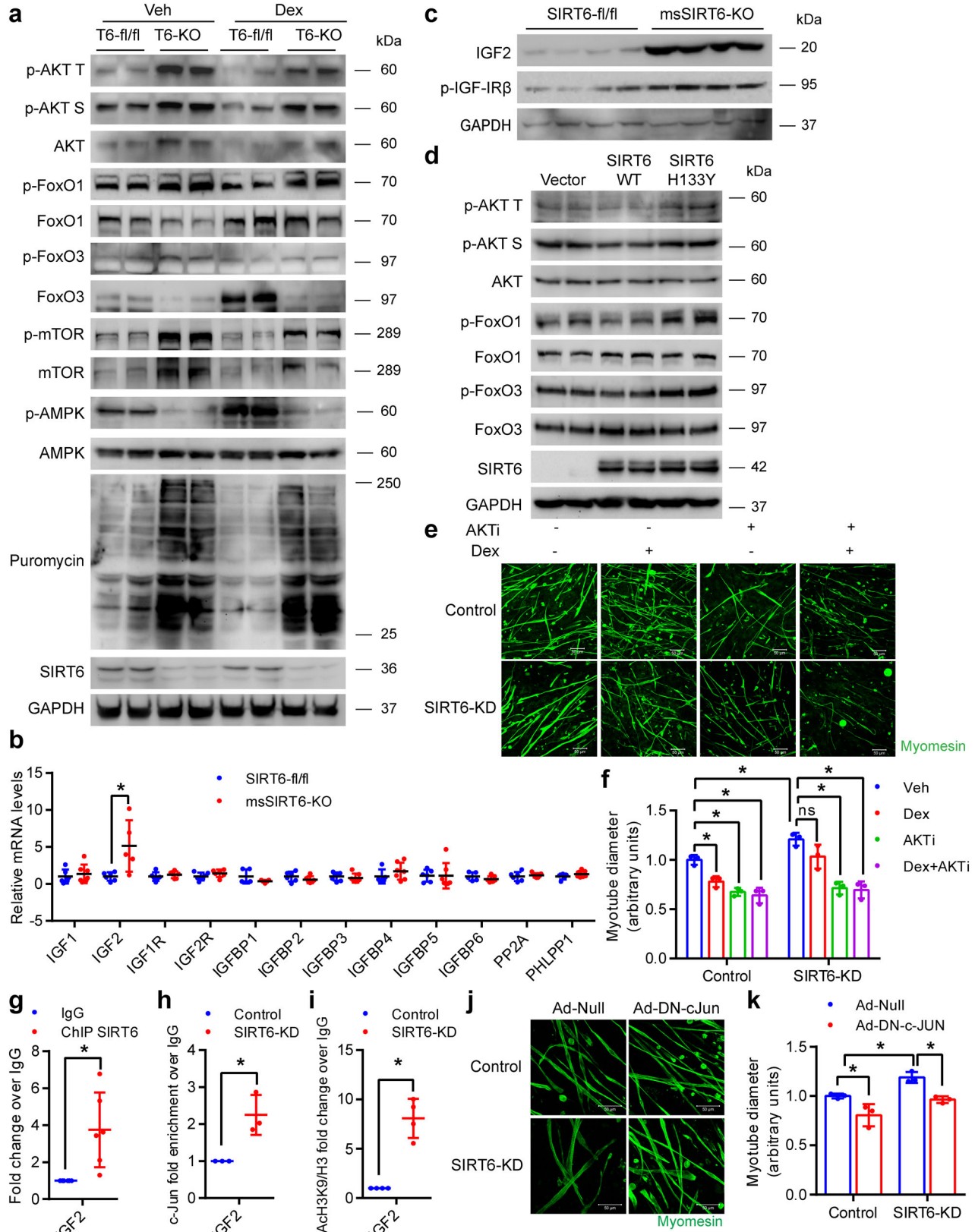

effects and/or adverse outcomes in other tissue types since SIRT6 is involved in a plethora of functions in various tissue. Therefore, usage of SIRT6 inhibitor should accompany suitable controls and further studies are needed to understand its effects on various tissues.

We exploited the germline-cre approach in generating two muscle-specific SIRT6-KO mice lines using ACTA or Myogenin cre

recombinase expressing mice and SIRT6-fl/fl mice. However, several complications are associated with this approach. The major complication of the germline-cre approach is the off-target effects that might confound results[61]. Studies indicate that transient ectopic expression of cre recombinase in the germline or at the early developmental stage could result in unexpected or unwanted recombination events. Such

**Fig. 9 | Muscle-specific SIRT6 deficient mice show hyperactive IGF/AKT signaling. a** Representative western blotting of AKT signaling proteins, phosphorylated protein levels of AMPK, total AMPK, and SUnSET analysis of protein synthesis in SIRT6-fl/fl and msSIRT6-KO mice gastrocnemius muscle after Dex or Veh injection for 15 days and quantifications are shown in Supplementary Fig. 16A–O. **b** qPCR analysis for expression of positive and negative regulators of IGF signaling in SIRT6-fl/fl and msSIRT6-KO mice muscle. $n$ = 5-7. **c** Representative western blotting of IGF2 and phosphorylated protein levels of IGF-I-Receptor β in SIRT6-fl/fl and msSIRT6-KO mice muscle and quantifications are shown in Supplementary Fig. 18A. **d** Western blotting images of phosphorylated and total protein levels of AKT, FoxO1, and FoxO3 in SIRT6 stable knockdown 293T cells transfected with pcDNA, SIRT6-WT, or SIRT6-H133Y plasmids and quantifications are shown in Supplementary Fig. 18B–E. **e** Representative confocal images showing myotube diameter in AKTi-1/2 and/or Dex-treated SIRT6 depleted myotubes. The myotubes were stained green with antibody against myomesin. Scale bar = 50 μm. **f** Quantification of myotube described in **e**. $n$ = 3. **g** ChIP analysis of SIRT6 binding to IGF2 promoter in C2C12 myotubes. $n$ = 6. **h** ChIP analysis of c-Jun binding to IGF2 promoter in control or SIRT6 depleted C2C12 myotubes. $n$ = 3. **i** ChIP analysis for acetylation of histone 3, lysine 9 residue at IGF2 promoter in control or SIRT6 depleted C2C12 myotubes. $n$ = 4. **j** Representative confocal images showing myotube diameter in SIRT6 depleted myotubes infected with either c-Jun dominant-negative (Ad-DN-c-Jun) or null adenovirus. The myotubes were stained green using an antibody against myomesin. Scale bar = 50 μm. **k** Quantification of myotube diameter described in **j**. $n$ = 3. Data presented as mean ± s.d., *$p$ < 0.05. Two-tailed unpaired Student's $t$-test was used was statistical analysis (**b**, **f–i**). Two-way ANOVA with Bonferroni post hoc test was used for statistical analysis (**k**). Dex (10 mg/kg/day) was administered (**a**). Source data are provided as a Source Data file.

recombination events might result in (1) partial or global recombination in all cell types, and (2) recombination might occur in cre-negative offspring indicating recombination events took place at the diploid stage of gametogenesis such that cre might not get segregated in haploid gametes[61]. Therefore, it is best to test such unexpected recombination events to avoid confounding results under such scenarios. But, we did not conduct any test in this respect which remains one of the limitations of this study. However, our western blotting result reveals that muscle-specific deletion of SIRT6 does not change SIRT6 protein levels in other tissue types (Supplementary Fig. 10B), suggesting that ACTA cre mediated deletion of SIRT6 has a comparably less off-target effect. Moreover, the conclusion of our study is derived from the experiments performed in two muscle-specific SIRT6 deleted mice lines. Though we observed overlapping muscle phenotype in both mice lines, the complications associated with cre lines listed above and other variables across the two mice lines might confound results. Therefore, the approach of comparing the results of two mice lines directly remains another limitation of the current study. In addition, the germ-line loss of SIRT6 raises another concern of secondary or tertiary changes/adaptation in the muscle as reported previously under prolonged AKT1/2 deficient condition leading to AMPK activation. In a similar line, we found changes in AMPK signaling in msSIRT6-KO mice muscle. However, the AMPK inhibitor was unable to rescue the phenotype in SIRT6 overexpressing conditions, suggesting changes in AMPK signaling occurs due to long-term loss of SIRT6 in muscle.

We found change in tibialis anterior and gastrocnemius muscle weight, but the weight of quadriceps, triceps, and biceps of Dex-treated msSIRT6-KO mice remained similar to control mice. Furthermore, the bodyweight of muscle-specific SIRT6 knock-out mice were comparable to control mice. The tibialis anterior and gastrocnemius muscle weight account for a very small fraction of the total weight of an animal. Therefore, we speculate that the change in only tibialis anterior and gastrocnemius weight will not significantly affect the total weight of the mice. To better understand the reason behind the discrepancy observed between molecular and phenotypic changes in msSIRT6-KO mice tibialis anterior muscle, we evaluated the proportion of fiber types by MHC staining. MHC fiber types in msSIRT6-KO mice were comparable to controls at the basal level. Interestingly, msSIRT6-KO mice tibialis anterior muscle was protected against Dex-induced fast to slow fiber type switch. Further, COX staining revealed larger fiber size in fast fibers in both ACTA cre and Myogenin cre-generated msSIRT6-KO mice line. Therefore, we speculate possible reasons as follows: (1) Increase in CSA might be fiber type-specific. However, we did not perform MHC staining and CSA together to draw any conclusion in this regard, (2) Increase in fiber number in msSIRT6-KO mice tibialis anterior muscle. Moreover, we measured CSA and MHC staining in the only tibialis anterior muscle of msSIRT6-KO mice, another limitation of this study. Though muscle mass accounts for a major portion of the body weight, we did not find any significant effect of muscle-specific deletion of SIRT6 on body weight. Therefore, we speculate that SIRT6 might not have a generalised action on all muscle types in the body, and further studies are needed to understand the muscle-type specific function of SIRT6.

Overall, the present study demonstrates that SIRT6 deficiency influences muscle homeostasis via transcriptional regulation of PI3K/AKT signaling.

## Methods
### Animal studies
Animal handling protocols were reviewed and duly approved by the Institutional Animal Ethics Committee (IAEC) of the Indian Institute of Science, constituted as per article number 13 of the Committee for the Purpose of Control and Supervision of Experiments on Animals (CPCSEA), Government of India. All the studies conformed with the Guide for the Care and Use of Laboratory Animals published by the US National Institute of Health (NIH Publication No. 85–23, revised 1996). Animals were housed in a pathogen-free facility with a standard chow diet in a 12 h light/dark cycle at Central Animal Facility with ad libitum food and water. SIRT6fl/fl (Sirt6tm1.1Cxd/J) and ACTA1-Cre (FVB.Cg-Tg(ACTA1-care)79Jme/J) mice used in the study were procured from the Jackson Laboratories, USA. Muscle-specific SIRT6 knockout mice were generated by crossing SIRT6-fl/fl mice with ACTA-Cre, SIRT6-fl/+ mice to obtain ACTA-Cre,SIRT6-fl/fl mice. All in vivo studies on msSIRT6-KO mice were performed on 4 months old male and female mice unless otherwise stated. The SIRT6-fl/fl littermates were used as controls for the experiment. Dex was administered intraperitoneally at 10 mg/kg/day for 15 days or otherwise stated. Comparable effects of Dex and SIRT6 deficiency on muscle phenotype were observed in both sexes. Mice tibia length was measured by a vernier caliper. Tibialis anterior, gastrocnemius muscle complex, quadriceps, biceps, triceps, and soleus muscles harvested from mice were weighed and snap-frozen immediately in liquid nitrogen. The absolute muscle weight and tibia length of mice used in this study are presented in tabular format in the Supplementary file. Supplementary Table 1 (Dex-induced muscle atrophy model), Supplementary Table 2 (msSIRT6-KO mice with/without Dex administration), and Supplementary Table 3 (SIRT6 inhibitor-treated mice with/without Dex administration). For the second line, C57BL/6-SIRT6 conditional mice (SIRT6fl/fl) that were previously generated[30] were crossed with C57BL/6 mice transgenic for the Cre recombinase under the control of the muscle-specific myogenin promoter (gift from P. Puigserver), Sirt6[F/+] to achieve deletion of SIRT6 specifically in muscle. Sirt6[F/F] Myogenin-cre mice were compared with the heterozygous Sirt6[F/+] Myogenin cre mice. The efficiency of the deletion was verified by PCR on DNA extracted from the skeletal muscle. All studies in this mice line were performed on 4 months old animals unless otherwise stated. Treadmill exhaustion test and energy expenditure analysis were carried out in 19 months old Myogenin-Cre mice. Four mice per genotype were analyzed using the treadmill and

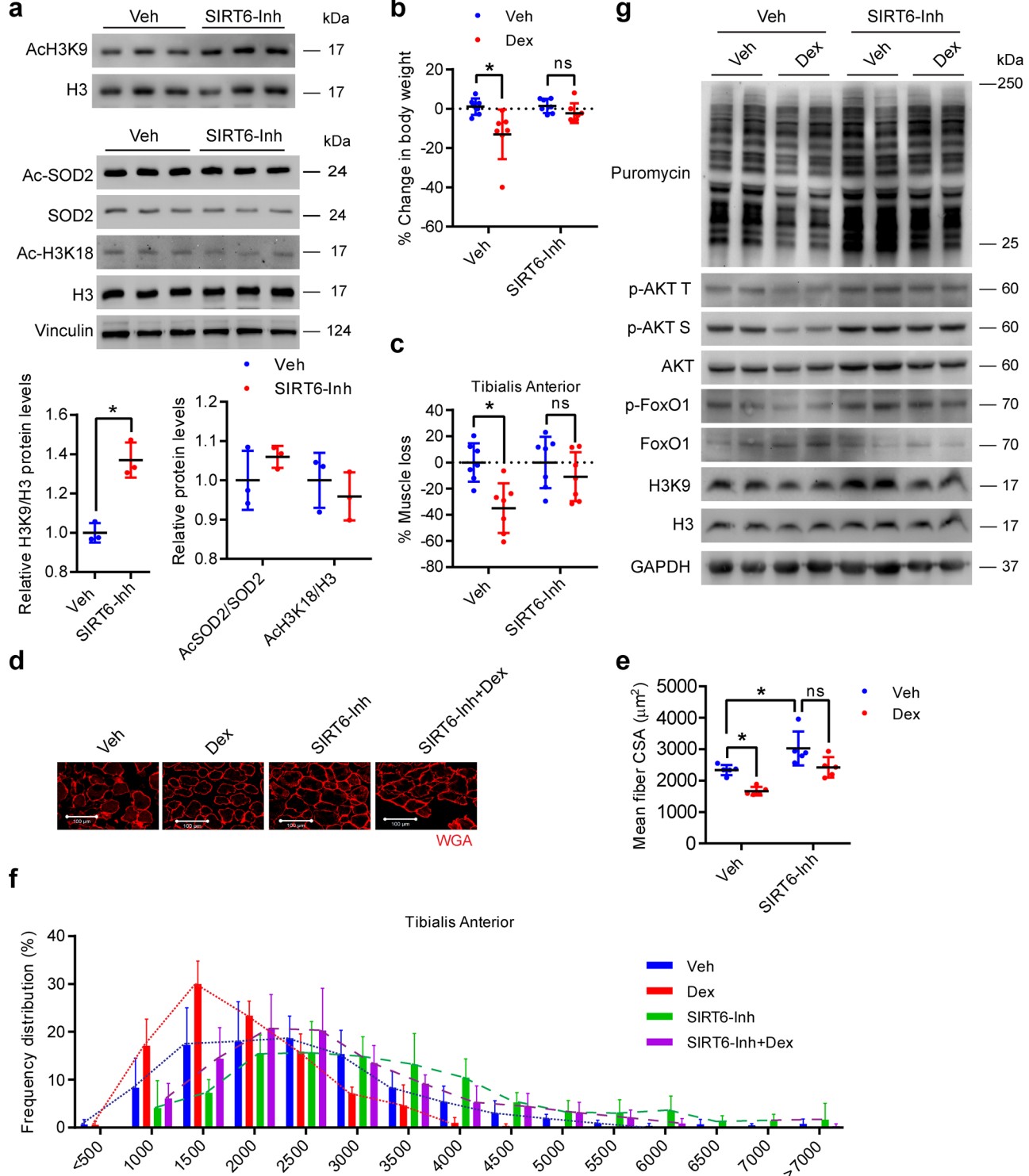

**Fig. 10 | SIRT6 inhibition protects mice against dexamethasone-induced muscle atrophy. a** Western blotting (top) and relative quantification (bottom) indicating the increase in global acetylation of histone 3 at lysine 9 in the gastro-cnemius muscle of SIRT6 inhibitor (SIRT6-Inh) administered mice. Western blot (middle) and relative quantification (bottom) of protein levels of acetylated SOD2 and global acetylation of histone 3 at lysine 18 normalized to SOD2 and histone 3 respectively in SIRT6-Inh treated mice soleus muscle. $n = 3$. **b** Change in body weight in SIRT6-Inh treated mice after Dex injection. $n = 7$. **c** Loss in TA muscle mass from SIRT6-Inh treated mice after Dex injection. $n = 7$. % of muscle loss in the Dex or SIRT6-Inh-Dex group is shown relative to Veh injected mice for the respective group. **d** Representative images showing WGA (red) stained SIRT6-Inh treated mice after Dex injection to determine the CSA. Scale bar = 100 μm. **e** Scatterplot showing mean fiber CSA in mice described in **d**. $n = 5$. **f** Frequency distribution of fiber cross-sectional area in mice described in **d**. $n = 5$. **g** Representative western blotting of AKT signaling, SUnSET analysis of protein synthesis, and global acetylation of histone 3 at lysine 9 in SIRT6-Inh administered mice with/without Dex treatment. Western blot quantifications are shown in Supplementary Fig. 21A–I. Data presented as mean ± s.d., ns not significant. Two-tailed unpaired Student's *t*-test was used for statistical analysis **a**. Two-way ANOVA with Bonferroni post hoc test was used for statistical analysis (**b**, **c**, **e**). Dex (20 mg/kg/day) and SIRT6-Inh (15 mg/kg/day) was administered. Source data are provided as a Source Data file.

indirect calorimetry system from TSE systems, as previously described[62]. Briefly, we used 0.25, 0.35 and 0.45 m/s speeds, for 300 s. To increase from one speed to the next, the system took 30 s. The shock grid was set at 1 mA for 0.5 s, and the experiment was stopped when mice touched the shock grid three times within 10 s.

For SIRT6 inhibitor experiments, the compound 2,4-dioxo-N-(4-(pyridin-3-yloxy)phenyl)−1,2,3,4-tetrahydroquinazoline-6-sulfonamide described previously[41,42] was administered at a dose of 15 mg/kg/day for 7 days with or without 20 mg/kg/day Dex. Peanut oil was used as a vehicle for SIRT6 inhibitor and PBS as a vehicle for Dex. C57BL/6J mice were used for these experiments.

For analysis of biochemical parameters, blood was collected from the retro-orbital sinus using a glass capillary and the serum was used for the tests. Estimation of triglyceride and total protein levels were performed using kits as per the manufacturer's protocol. For measuring fasting blood glucose level, blood was drawn from the tail vein and the glucose level was measured using a commercially available glucometer according to standard protocol.

## Muscle histology and immunohistochemistry

Gastrocnemius muscle was collected and immediately fixed in 10% neutral buffered formalin for histological analysis. An automated tissue processor was used to process the muscle tissue before histological analysis. H&E and Masson's Trichrome staining was performed according to standard procedures to observe degenerating muscle and to determine the extent of fibrosis in SIRT6-fl/fl and msSIRT6-KO skeletal muscles. Animals were randomly selected from each group for H&E and fibrosis analysis. Fibrosis was analyzed manually across the entire tissue section and multiple non-overlapping frames of the entire tissue section were captured from longitudinal sections of the muscle. Three sections per tissue were stained and analyzed, while the observer was blinded to the experimental groups. Fibrosis scoring was done using Trichrome stained sections in a blinded fashion on a scale of 1–5, with 5 indicating maximum fibrosis.

CSA analysis and MHC fiber typing were performed in the tibialis anterior, gastrocnemius, soleus, quadriceps, triceps, and biceps muscles as described previously with a few modifications[63]. Muscles were immersed in a PolyFreeze medium and frozen using liquid N$_2$-chilled isopentane. We selected animals with highest, lowest end and middle muscle weight for (1) mice administrated with Dex, and (2) msSIRT6-KO administrated with Dex. However, five animals were randomly selected from each group for SIRT6 inhibitor-treated mice with/without Dex. Tissue blocks were cryosectioned at 5 μm thickness using a cryo-microtome at −20 °C and serial sections were stained with wheat germ agglutinin (WGA) Alexa Fluor™ 594 conjugate or antibodies against MHC I or cocktail of MHC IIA MHC IIB or MHC IIX. Briefly, muscle sections were fixed with 4% paraformaldehyde and blocked with 1% BSA solution. One section per tissue was stained with primary antibodies against MHC I (BAD-5, DSHB), MHC IIA (SC-71, DSHB), MHC IIB (BF-F3, DSHB), and MHC IIX (6H1) followed by secondary antibodies: Goat anti-Mouse IgG Secondary Antibody, Alexa Fluor 488 against MHC I, Goat anti-Mouse IgG1 Cross-Adsorbed Secondary Antibody, Alexa Fluor 594 against MHC IIA, and Goat anti-Mouse IgM (Heavy chain) Cross- Adsorbed Secondary Antibody, Alexa Fluor 488 against MHC IIB and MHC IIX. Sections were mounted with Fluoromount using clean coverslips. Images were captured in a similar manner after WGA staining for the determination of myofiber CSA. Tile scan confocal images were acquired at 10x followed by stitching to generate whole tissue images using Zeiss LSM 880 upright confocal multiphoton microscope. Percentage (%) of each MHC I, MHC IIA, MHC IIB, and MHC IIX positive fibers to the total cross-sectional fibers were calculated manually using ImageJ software as described previously[64]. CSA of a minimum of 200 individual cross-sectional myofibers per muscle or the maximum possible number of cross-sectional myofiber were manually analyzed for each muscle using ImageJ software, while the observer was blinded to experimental groups. Cytochrome oxidase (COX) was performed as described previously[65,66] to determine the proportion of oxidative and glycolytic fibers. Briefly, after drying the slide containing a 6 μm thick muscle section (cut using a cryostat at −20 °C) for 30 min, the slides were incubated at 37 °C for 2 h in a solution containing 1xDAB, 100 μM cytochrome C in 0.1 M PBS, pH 7.2 and 2 μg/ml bovine catalase. The slides were washed with 0.1 M PBS (pH7.0) four times for 10 min each. The slides were then washed in deionized water for 3 min, dehydrated in 50% ethanol for 2 min, and were mounted with DPX. Pictures were taken at 20X.

For the second line of mice (SIRT6fl/fl Myogenin Cre), soleus, and gastrocnemius muscles from 4 months old F/+ MyoCre and F/F MyoCre mice were collected in OCT, and cross-sections were stained with H&E to determine the fiber diameter. The diameter of the myofibers in the cross-section was measured in ImageJ and the data from the different genotypes were compared by unpaired two-tailed $t$-test in Prism.

## Glucose tolerance test (GTT) and weight

One-month-old mice were fasted overnight. In the morning the weight and basal blood glucose were determined. 2 g/Kg of glucose was then administered by intraperitoneal injection. Blood glucose level was determined again at 30', 60', 90', and 120' after injection. Blood glucose was measured with Accu-Chek test strips (Aviva).

## Cell culture and primary myotube culture

293T and C2C12 cell lines were procured from ATCC. All cells were maintained at 37 °C and 5% CO$_2$ in DMEM supplemented with 10% fetal bovine serum, 100 units/ml penicillin, and 100 μg/ml streptomycin. Transfections were carried out using Lipofectamine RNAiMAX for siRNA or Lipofectamine 2000 for plasmids for 72 h, as per the manufacturer's protocol. For adenovirus-mediated transduction, primary myotubes were infected with the adenovirus for 72 h.

Murine primary myotubes were cultured as described previously[67]. Briefly, muscle tissue from 0–2 days old pups were digested in an enzymatic mixture consisting of 0.2% trypsin, 0.4 mg/ml Collagenase Type II and 0.01 M of D-glucose in PBS. The digestion was carried out in shaking incubator for 10 min at 37 °C and was repeated for ten cycles. The supernatant from the first digestion containing the debris was discarded. The single-cell suspension from the subsequent digestion steps were collected in 100% horse serum and maintained at 37 °C. The suspension was pre-plated for 45 min to remove cells of non-myoblast lineage. The supernatant enriched in the myoblast population was then collected by centrifugation and seeded in tissue culture plates or sterilized coverslips in high glucose Dulbecco's Modified Eagle Medium (DMEM, Sigma-Aldrich) supplemented with 10% fetal bovine serum,100 units/ml penicillin and 100 μg/ml streptomycin. Murine myoblasts were differentiated to myotubes in DMEM supplemented with 4% horse serum, 100 units/ml penicillin, and 100 μg/ml streptomycin. The media was changed to high glucose DMEM with 10% fetal bovine serum, 100 units/ml penicillin, and 100 μg/ml streptomycin after differentiation. For induction of myotube diameter reduction, murine primary myotubes were treated with 50 μM Dex for 24 or 48 h. For inhibition of AKT activity, primary myotubes were treated with either 1 μM AKTi-1/2-or 10 μM LY294002 for 24 h. For inhibition of AMPK activity, primary myotubes were treated with 10 μM Compound C for 24 h. The average myotube diameter was calculated as described previously[68]. Furthermore, protein synthesis rates in myotubes were assessed using the SUnSET assay as described in our previous study[69]. The relative puromycin levels were measured after normalization with either GAPDH or ponceau.

## Western blotting

Lysates were prepared from cultured cells, primary myotubes or mouse gastrocnemius muscle using lysis buffer containing 1 mM EDTA, 1 mM EGTA, 50 mM tris-Cl pH 7.8, 1% NP40, 150 mM NaCl, 0.5% sodium deoxycholate and 1% sodium dodecyl sulfate supplemented with sodium ortho-vanadate, sodium pyrophosphate, protease inhibitor cocktail, and PMSF. Western blotting was then performed as described previously[33]. Briefly, the proteins were quantified by the Bradford assay and equal amounts of protein were loaded onto SDS-PAGE gels and resolved by electrophoresis. The proteins were then transferred to PVDF membranes, blocked with 5% milk, and then probed with specific antibodies.

The activity of SIRT3 and SIRT7 were analyzed in the mice soleus muscle treated with SIRT6 inhibitor or Veh. Muscle was homogenized in the lysis buffer containing 50 mM Tris-HCl pH 8, 150 mM NaCl, 1 mM EDTA, 1 mM NaF, 10 μM trichostatin A, 10 mM nicotinamide, 0.5 mM DTT, and protease inhibitor cocktail. 30 μg of protein from muscle lysate was loaded on a 10% polyacrylamide gel and separated by SDS-PAGE. Proteins were subsequently transferred to nitrocellulose membranes and probed with specific primary antibodies.

## Confocal microscopy

Immunofluorescence was performed as described previously[33]. Briefly, cells were fixed with 3.7% formaldehyde, permeabilized with 0.2% Triton X-100, and then blocked with 5% BSA in PBST. The cells were then incubated with the primary antibodies overnight. After washing in PBS, cells were incubated with secondary antibodies conjugated with Alexa fluor 488 and/or 546/594. Hoechst 33342 was used to stain the nucleus. Coverslips were mounted on a clean glass slide using Fluoromount and images were acquired using a confocal microscope.

## Chromatin Immunoprecipitation

ChIP was performed from mice muscles or the C2C12 cell line using the commercial kit (Cell Signaling) as per the manufacturer's protocol. Briefly, cells or tissues were first cross-linked with 1% formaldehyde for 10 min and then quenched in 0.125 M glycine for 5 min. The chromatin was then sheared using micrococcal nuclease treatment followed by sonication. Chromatin sheared in the range of 500–1000 bp was used for immunoprecipitating DNA-protein complex. Normal rabbit IgG antibody was used as control. DNA-protein complex was then decrosslinked overnight and following DNA isolation, qPCR was performed using specific primers.

## Real-time qPCR analysis

qPCR was performed as described in our previous study[33]. Briefly, RNA was isolated using the Trizol reagent as per standard protocol. The RNA was then quantified by nanodrop and equal amounts of RNA were used for cDNA synthesis by reverse transcriptase, as per the manufacturer's protocol. qPCR was performed using SYBR-Green master mix and specific primers in a real-time PCR system.

## Amino acids analysis using reverse-phase HPLC

Reverse-phase liquid chromatography was used to separate and quantify 6-Aminoquinolyl-N-hydroxysuccinimidyl Carbamate (AQC) derivatized amino acids as per previous publication[70]. Briefly, 250 μl of total cell lysate was deproteinized by mixing with 166.6 μL of acetonitrile (60:40 v/v) and centrifuged for 1 min. Following centrifugation, the supernatant was mixed with 1 μL (10 mg/ml Norleucine− Internal standard), 0.6 μL of 20 mM HCl, 160 μL 0.2 M Borate buffer (pH 8.8), and 40 μL AQC (3 mg/ml) and heated at 50 °C for 10 min. The mixture was filtered through 0.2 micron filter and kept for HPLC injection. HPLC analysis was performed using Shimadzu, UPLC LC-20 series system with auto-sample injection. The column for the HPLC

separation was a $C_{18}$ column (150 × 4.6 mm I.D., 3 μm) (Inertsil-ODS-2) thermostated at 37 °C with a simple multistep linear gradient of two solvents at a flow rate of 1 mL/min. Solvent A was 140 mM Sodium acetate + 17 mM Triethylamine (pH 5.05), and solvent B was water-acetonitrile (40:60, v/v). The gradient profile for the HPLC setting is provided in Supplementary Table 4. The detection of the separated AQC amino acid derivatives was done using a UV detector at 254 nm. The efficacy of the protocol was validated by running an amino acid standard cocktail (Catalogue No. AA-S-18, Sigma). The area under the curve (AUC) of the respective amino acid peaks was normalized against norleucine AUC and further normalized with DNA content. Qubit™ dsDNA HS Assay Kit was used for DNA quantification as per the manufacturer's protocol.

## Luciferase assay

FoxO reporter assay was performed using FHRE-Luc plasmid (firefly luciferase) in primary myotubes as previously described[71]. A Renilla-Luc plasmid (pRL-CMV) was co-transfected and used for normalizing luciferase readings.

## Quantification, statistical analysis, and software

Data analysis and graph preparation were performed using Graph-pad prism (Version 6.04). Student's $t$-test, one-way ANOVA or two-way ANOVA was used to perform pairwise comparisons and multiple comparisons with the Bonferroni post hoc test. A $p$ value of less than 0.05 was considered significant. $P$ value and ANOVA table analysis are provided in Supplementary Data 1. Confocal images were processed using Zen 2.1 Black software and quantification was done using ImageJ version 1.53c software. ImageJ was used for quantification of myotube diameter, the cross-sectional area of muscle fibers, fiber typing, mean fluorescence intensity and nuclear localization of FoxO1 and FoxO3 and densitometrical analysis of western blots.

## Key resources table

The details of antibodies used in the study is presented in Supplementary Table 5 and details of primers, siRNAs, plasmids, adenovirus, kits, chemicals, reagents, equipments used in the study are provided in Supplementary Data 2.

## Reporting summary

Further information on research design is available in the Nature Research Reporting Summary linked to this article.

## Data availability

Source data supporting the findings in this study underlying dot plots, bar graphs, and uncropped western blots are provided in the online source data file. Source data are provided with this paper.

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

## Acknowledgements

We acknowledge the Central Animal Facility, Central Confocal facility (Division of Biological Sciences and Department of Microbiology and Cell Biology) and the microtome facility (Division of Biological Sciences) all at IISc, Bengaluru for their services and the technical help. We thank P.A.D. for technical advice in histology and immunohistochemistry experiments for SIRT6-ACTA Cre mice. We also thank P. Puigserver for the Myogenin-Cre mice; Lei Zhong, Carlos Sebastian and Liping Zhao for technical advice; and the Histopathology Core of the A. Einstein College of Medicine for processing samples from the SIRT6-Myogenin Cre mice. N.R.S. acknowledges funding from the Department of Biotechnology, Government of India, the Department of Science and Technology Extra Mural Research Funding, the Department of Science and Technology- Fund for Improvement of S&T Infrastructure, Government of India, and the Department of Biotechnology–Indian Institute of Science partnership program for advanced research. E.A. acknowledges funding from European Union's Horizon 2020 research and innovation programme under the Marie Skłodowska-Curie grant agreement No 671881 (INTEGRATA). R.M. acknowledges funding from NIH grants (R01GM128448 and R33ES025638).

## Author contributions

N.R.S. conceived and designed the study. S.M. and V.R. analyzed results and wrote the manuscript. D.K. generated ACTA-cre msSIRT6-KO mice. S.M., A.K.T., D.K., and C.C. performed experiments in animals. S.M., A.J., and A.K.T. cultured primary myotubes. S.M. and A.B.P. performed confocal microscopy. S.M. performed luciferase assay. S. Kumar and D.K. performed ChIP experiments. S.M., V.R., S.S., E.A., and A.B.P. performed western blotting. S.M., D.K., A.K.T., S.M.K., S.R., and S. Krishna performed qPCR analysis. S.M., S.S., N.A.J., and A.K.T. performed histology and immunohistochemistry. A.S.R. and T.J. performed HPLC analysis. D.K. and R.R. performed genotyping. C.C., E.R.H., I.J.K., and R.M. generated the second conditional KO line of mice and performed analysis on the muscle of these animals. S.B. and E.A. provided the inhibitor and reviewed the manuscript. R.P.V. was involved in the animal experiments. N.R.S. and R.M. interpreted the data, coordinated the study, and prepared the final version of the manuscript.

## Competing interests

R.M. has a financial interest in Galilei Biosciences, a company developing activators of the mammalian SIRT6 protein. R.M.'s interests were reviewed and are managed by MGH and MGB Healthcare in accordance with their conflict-of-interest policies. The other authors declare no competing interests.

## Additional information

Sneha Mishra[1,10], Claudia Cosentino[2,10], Ankit Kumar Tamta[1], Danish Khan[1], Shalini Srinivasan[1], Venkatraman Ravi[1], Elena Abbotto ®[3], Bangalore Prabhashankar Arathi[1], Shweta Kumar[1], Aditi Jain[4], Anand S. Ramaian[5], Shruti M. Kizkekra[1], Raksha Rajagopal[1], Swathi Rao[1], Swati Krishna[1], Ninitha Asirvatham-Jeyaraj[6], Elizabeth R. Haggerty[2], Dafne M. Silberman[7], Irwin J. Kurland[8], Ravindra P. Veeranna[9], Tamilselvan Jayavelu[5], Santina Bruzzone[3], Raul Mostoslavsky ®[2] ✉ & Nagalingam R. Sundaresan[1] ✉

[1]Department of Microbiology and Cell Biology, Indian Institute of Science, Bengaluru, India. [2]The Massachusetts General Hospital Cancer Center, Harvard Medical School, Boston, MA, USA. [3]Department of Experimental Medicine, Section of Biochemistry, University of Genoa, Genoa, Italy. [4]Centre for BioSystems Science and Engineering, Indian Institute of Science, Bengaluru, India. [5]Department of Biotechnology, Anna University, Chennai, India. [6]Department of Biotechnology, Indian Institute of Technology, Chennai, India. [7]Centro de Estudios Farmacologicos y Botanicos (CEFYBO-CONICET), Catedra de Farmacologia, Facultad de Medicina, UBA, Buenos Aires, Argentina. [8]Albert Einstein College of Medicine, Bronx, NY, USA. [9]Department of Biochemistry, CSIR-Central Food Technological Research Institute, Mysuru, India. [10]These authors contributed equally: Sneha Mishra, Claudia Cosentino. ✉e-mail: rmostoslavsky@mgh.harvard.edu; rsundaresan@iisc.ac.in

