## [Peer Review File · Nature Communications]

Reviewers' Comments:

Reviewer #1:

Remarks to the Author:

The authors present a novel research study, examining whether activation of the NAD-dependent deacetylase Sirtuin 6 (SIRT6) is a molecular regulator of glucocorticoid-induced skeletal muscle atrophy. They report the following claims: (1) SIRT6 is rapidly increased in skeletal muscle following dexamethasone (DEX) treatment, (2) genetic induction of SIRT6 results in muscle atrophy, (3) genetic ablation of SIRT6 protects from DEX-induced muscle atrophy, (4) pharmaceutical inhibition of SIRT6 protects from DEX-induced muscle atrophy, and (5), SIRT6 appears to convey its regulatory role through an IGF2-AKT signalling pathway.

Overall the work is of great interest and addresses a novel research angle, given that there is little known about the role(s) of Sirtuins in the regulation of skeletal muscle mass. There is considerable interest in therapeutic approaches to prevent muscle wasting, and so the muscle-sparing effect of the SIRT6 inhibitor would be of interest to the wider-research community.

The use of muscle-specific loss and gain of function SIRT6 mice, viral SIRT6 deletion in vitro and the use of a pharmaceutical inhibitor of SIRT6 in vivo provides convincing data to support the authors claims. I have some questions regarding the analysis performed that are listed below.

There is little information known about Sirtuin action in skeletal muscle, and considerable focus on Sirtuin activators for health benefit. Therefore, describing a muscle centric role for SIRT6 as the authors have done, in addition to describing a therapeutic role for SIRT6 inhibition could lead to new direction of thinking in the field.

Specific comments:

1. There is a lack of consistency across the various in vivo models with regard to the analysis approaches performed. If quantifying muscle atrophy, the authors should always report (1) muscle weight, (2) CSA and (3) fibre distribution. They have done this in the SIRT6 KO mice (Figure 3), but not for DEX-induced atrophy (Figure 1) or the inhibitor study (Figure 6). Therefore, please add this data in Figures 1 and 6.
2. All SUnSET data needs to be quantified and reported as group data, rather than just presented as a western blot image. Please provide corresponding group data for Figures 1H, 2F, 2K and 6H.
3. All western blot signalling data needs to be quantified and reported as group data, especially the phospho/total data presented (Figure 5A, 5C, 5D, 6A) and the phospho/acetyl regulatory sites listed on the figure.
4. Protein content of all other Sirtuins should be measured in the SIRT6 KO and SIRT6 Tg mice to complement the mRNA data reported.
5. Is there a fibre-type shift observed in the SIRT6 KO and Tg mice? COX staining does not achieve this on its own and needs to be complemented with MHC staining.
6. Is there a change in habitual energy expenditure or muscle-specific mitochondrial respiration in SIRT6 KO and Tg mice? Both seem important given that the authors are assessing chronic muscle atrophy in which daily activity/mito function will dramatically alter the results.
7. The authors suggest that the lack of effect of Dex on muscle atrophy in the Quadriceps, Triceps and Biceps (compared to GTN or TA) is due to differing proportion of fast twitch fibres (Figure 1). Please support this with analysis of fibre-type profiles in these muscles. Based on this observation, it is also unclear why the authors did not examine the Soleus muscle?
8. The authors need to demonstrate the specificity of the Atrogin-1, MuRF1 and FOXO3a antibodies they used in their IHC analysis as the majority of the positive staining presented doesn't appear to actually be in myotubes (rather in non-differentiated cells). This can be achieved by staining following Atrogin-1, MuRF1 or FOXO3a viral silencing (or peptide competition assays) as they have already performed for SIRT6.
9. It would be useful to the reader if the authors also provided the muscle weight data as a % decline, as they have done for body weight (Figure 1B, 3C-G, 6C-G).
10. Specificity of the SIRT6 inhibitor needs to be demonstrated in vivo. Please report changes in SIRT1-7 mRNA, protein and activity for each (Figure 6). The authors should also repeat the Ach3K9 analysis pre and post DEX intervention (i.e. add to the analysis already performed in

Figure 6H.

11. The authors generate their SIRT6 models using different Cre lines, but then compare the two models directly. This seems problematic and I think needs to be further highlighted and discussed as a limitation to the data interpretation.

12. If the authors are proposing that SIRT6 is driving its biological action via modulation of AKT activity, then it seems pertinent to directly test this through an Akt inhibitor, rather than a PI3K inhibitor. Indeed, LY294002 has been shown to have numerous off target effects (Bain et al PMID: 17850214) that limit interpretation using this compound.

13. The authors suggest that FOXO proteins are regulating some of the biological effects shown, and report data on FOXO1 and FOXO3. It is unclear if they are proposing both to play a role, as in one model they report FOXO3 data (Figure 4) and another FOXO1 data (Figure 5). Can these proteins be used interchangeably? Presumably they are not regulated by universal processes?

14. Introduction – The authors discuss protein balance and muscle wasting (line 4), however I don't think it is possible to group ageing and the other disorders together as they have as the mechanisms of muscle atrophy are different. Sarcopenia is thought to be driven by a reduction in protein synthesis (not elevations in degradation), whereas the other inflammatory conditions listed are probably a combination of changes in synthesis and degradation.

Reviewer #2:

Remarks to the Author:

Using primary myotubes and a muscle-specific SIRT6 KO mouse (which was generated by the authors), Mishra and colleagues investigated the role of SIRT6 in dexamethasone-induced muscle atrophy. In informative cell-based and muscle-based analysis they found that modulating SIRT6 expression altered the susceptibility to dexamethasone-induced muscle atrophy, such that KO of SIRT6 was protective. Additionally, they used some nice molecular analysis to inform mechanism of action, and concluded that SIRT6 works through modifying FoxO, c-Jun, IGF2 and PI3K/Akt signaling.

Major comments:

Muscle fibre area is difficult to measure accurately. The approach used by the authors of formalin fixing makes it very difficult to accurately assess muscle fibre area, as the muscle is not pinned at a consistent length. Further to this point, by not pinning the muscles it is very difficult to make an accurate measure of CSA, and it is evident from the WGA images that the fibre sections (at least for some) are cut at an oblique angle. This makes the measurement of fibre area inaccurate. Overall, given the importance of muscle fibre area to the premise of the paper this point should be addressed.

Assessing fibrosis etc. is tricky. More details on the stereological approach is needed. How many sections? What area of the muscle? How much of the muscle cross section was assessed? Was it randomised? An unbiased approach (blinding is mentioned in the Fig legend) is important for accurate measure of histological samples. This applies to the muscle CSA analysis as well.

The muscle fibres of the mouse are pretty much all "fast" (e.g. PMID: 7932768, 22938020). For example, the quadriceps, TA, triceps, biceps and gastrocnemius are all similar (i.e. predominantly "fast" type 2). The argument that the quadriceps, triceps and biceps did not decrease in weight with glucocorticoid treatment due to fibre type is not accurate. This point should be addressed.

With Dex, was SIRT6 expression differentially effected/increased in the gastrocnemius versus the quadriceps/triceps/biceps? Given muscle weight was differentially effected in these muscles with Dex, per the premise of the study, it would be interesting/important to know if SIRT6 expression was also differentially effected. This should be measured/presented. In fact, this is a very interesting aspect of the study. Why there is a different response in different muscles and how does this fit with the author's findings and premise?

It is not clear from Figure 1 if SIRT6 increased after or before the atrophy occurred. This point should be made. A nice addition to the paper would be temporal changes in muscle weight and

SIRT6 expression over the course of the 15 days of Dex. Regardless, the point above will help address this point.

Quantification of all western and puromycin blots should be provided throughout the manuscript. Also, for the puromycin blots, did the authors run a ponceau stain (this is the common approach)? That would help verify equal loading, which is critical for such an assay.

Figure 3B-G. The number of mice on which body weight was measured is different from the number of mice for which there is muscle weights. It is not clear why this is the case.

Figure 3B-G. Is the data assessed by 2 way ANOVA? If so, was there a main effect for Dex? Main effects, interaction and post-hoc analysis should be presented. This data is important to the studies' premise so providing such data will be informative to the reader. The type of analysis should be provided for all figures with similar analysis (that is, 2 x 2 design).

In the msSIRT6KO mice, was SIRT6 comparably knocked out across all muscles?

For Dex treatment of mice, rather than % change in body weight it would be helpful to show the pre/post body weight for each mouse (with lines joining them). This could be a Suppl figure, if needed.

Are the analyses performed in male or female mice? If one sex, are the findings of primary measures consistent in the other sex? Please update Methods with details.

The authors measured muscle weight on 7-10 mice, yet the fibre area is on only 4 mice. How did the authors select which mice to run CSA measures on? Was this randomized? CSA data for all mice on which muscle weight was analysed should be presented. At the very least, a distribution of mice that represents the high, middle and low end of muscle weight would be helpful.

The breeding strategy for the msSIRT6KO mice is unclear. Are the fl/fl mice, littermate controls? Or, were 2 separate lines of mice bred to provide the Cre+ and Cre- mice? Please provide details.

Figure 3L. What are the results of the 2 way ANOVA analysis for this figure? Are there differences?

Related to this statement: This sentence does not refer to the correct data: However, we also observe myofiber enlargement at basal conditions in these animals, further validating a role for SIRT6 in modulating fiber size (Fig. S3D-E). This data is a combination of two different muscles? This is not an appropriate representation of the data. It is also not appropriate to present individual fibre data. Please separate the two muscle types and just present the average fibre area.

FoxO transcriptional activity is reduced ~50% in SIRT6-KD cells. Thus, there is still 50% activity. Do the authors think that there is a 'tipping' point below which FoxO activity is sufficiently reduced such that it is protective? Physiologically speaking, does this seem reasonable?

Please provide quantification of the blots in Fig 5. How long was Dex administered for in this data (please state in text of Fig Legend)?

Also, in Fig 5A, SIRT6 protein did not increase (looks like it decreases?) with Dex in fl/fl mice. This does not fit with the model proposed by the authors, nor the data in Figure 1. Some comment in the text is needed.

Please provide quantification of Figs 6A and 6H. What is the n for these experiments?

Similar to above, for Fig 6B-G, please provide the results of the 2 way ANOVA table analysis in the figure.

The muscle-specific SIRT6 KO mice is a nice model. However, the authors used a germline Cre approach. The limitations of this approach should also be addressed/acknowledged.

The relevance of when Cre turns on during development in MCK vs HSA Cre mice, as is discussed in the Discussion, is unclear. Why does it matter when it turns on in terms of effects on measures that are made?

The 3rd paragraph of the Discussion does not seem relevant to the paper. Or, could be moved toward the end of the Discussion if it is felt that it adds value.

Do the authors think that their mechanism works through Akt1? Some discussion on Akt isoforms in muscle is warranted.

Given SIRT6 KO in muscle has previously been shown to effect AMPK activity, do the authors find any effect on AMPK in their model (e.g. pAMPK or pACC)? This would be a nice inclusion to the paper.

Much of the analysis has an n=3 or 4 mice. This is very low for mouse studies, especially of muscle biology. What is the rationale for this?

Absolute (i.e. not corrected for tibia length) muscle weights should be presented in a table. This could go in the Suppl Files. Given that the authors find no effects on body weight then the absolute data will be nice to see. As an aside, was tibia length different between mouse models?

Was total muscle protein concentration differentially effected in Dex and SIRT6 KO mice? This would be a nice addition related to atrophy.

Was protein synthesis effected in msSIRT6 KO mice? Such analysis would tie in nicely with the differences in fibre area etc.

Minor comments:

For the gastrocnemius weights, was this the gastrocnemius complex, or was the plantaris muscle dissected out? Please provide details.

In Figure 1J, the SIRT6 blot looks like a spliced blot. If this is the case, please mark the blot as such.

The post-hoc test used for ANOVA analyses is not mentioned.

It would help the reader if the y-axis (or above the figure) had more information about the measure being made. Otherwise, one needs to keep referring to the text or Fig Legend.

Reviewer #3:

Remarks to the Author:

The paper by Mishra and collaborators investigates the role played by sirtuin 6 (SIRT6) in dexamethasone (DEX) induced skeletal muscle atrophy. The study combines both in vitro and in vivo approaches and shows that, while SIRT6 overexpression is associated with muscle atrophy, SIRT6 depletion plays a protective role. The following points should be addressed and improved:

- Results, page 4: the authors claim that they have established an experimental model of muscle atrophy based on DEX administration. However, this model is widely used and very well characterized since long ago. In this regard, the only novelty is the up-regulation of SIRT6 expression. I would encourage the authors to clearly indicate in the text that the data reported in Figure 1, panels A-H, are just confirmatory;
- Results page 4: the authors state that they validated the in vivo data with results obtained in C2C12 cultures exposed to DEX. How can cell cultures be used to validate an in vivo study?
- the definition 'myotube atrophy' is not appropriate, better saying 'reduction/decrease in size', 'smaller than', etc.;

- SIRT6 KO mice appear comparable to controls, in particular no changes in body weight and muscle mass are reported. However, muscle cross sectional area (CSA) is increased while the expression of molecules involved in the proteolytic machinery is reduced, suggesting that the lack of SIRT6 affects muscle homeostasis also in the absence of a catabolic stimulus (in this case, DEX). How do the authors explain that such effects are not reflected on muscle mass?
- strictly related to the previous point, all the in vivo staff lacks very important information on muscle function. While it is known that DEX impinges also on muscle contractile activity, nothing is demonstrated in the present study when SIRT6 is manipulated (both deleted and overexpressed);
- the authors justify the striking discrepancy among their data and previously published results (ref 34 in the paper), discussing that while these latter were obtained using a total body SIRT6 KO, their model relies on a muscle-specific deletion. This is totally reasonable, however, the authors obtain protection against muscle atrophy also using a pharmacological SIRT6 inhibitor, that is practically mimicking the total body KO. How do they explain such a different pattern?
- Figure 1, panel B: please check point distribution for the Triceps muscle;
- immunofluorescence experiments should be confirmed by western blotting;
- DEX effect on protein synthesis is very clear in Figure 1H but not in Figures 2K and 6H;
- the effect of SIRT6 inhibitor should also be evaluated on myofiber CSA, the more so since SIRT6 genetic ablation results in basal CSA enlargement;
- Figure 5A, 5B and 5C, empty vector should be shown as well.

RESPONSE TO REVIEWERS' COMMENTS

We thank the reviewers for the insightful comments and suggestions that has helped us to tremendously improve the quality of our work. These comments have provided us the opportunity to understand the role of SIRT6 in muscle biology, a relatively less explored area in sirtuin biology. We have put our best efforts in addressing reviewers concerns and have provided point-to-point response to reviewer's comments. All changes in the manuscript are highlighted in yellow.

Reviewer #1 comments

Comment 1: There is a lack of consistency across the various in vivo models with regard to the analysis approaches performed. If quantifying muscle atrophy, the authors should always report (1) muscle weight, (2) CSA and (3) fibre distribution. They have done this in the SIRT6 KO mice (Figure 3), but not for DEX-induced atrophy (Figure 1) or the inhibitor study (Figure 6). Therefore, please add this data in Figures 1 and 6.

Response: We thank the reviewer for the insightful suggestion. We performed analysis as per suggestion and have included: (1) muscle weight, (2) CSA, and (3) fibre distribution for the following mice models: (1) Mice administered with dexamethasone (Dex), (2) muscle-specific SIRT6-KO mice with/without Dex, (3) SIRT6-inhibitor treated mice with/without Dex. Please see Figure 1B-1E, Supplementary figure 1-6, Figure 4C, 4D, 4F, 4G, Supplementary figure 13, Figure 10C, 10D-5F, Supplementary figure 18. Page 4, 5, 7, 8, 13.

Comment 2: All SUnSET data needs to be quantified and reported as group data, rather than just presented as a western blot image. Please provide corresponding group data for Figures 1H, 2F, 2K and 6H.

Response: As per the suggestion, we have quantified all SUnSET assays and represented in the form of group data. Please see Figure 1K, 2F, 3F, 9A, Supplementary figure 15L, 10G, Supplementary figure 19A.

Comment 3: All western blot signalling data needs to be quantified and reported as group data, especially the phospho/total data presented (Figure 5A, 5C, 5D, 6A) and the phospho/acetyl regulatory sites listed on the figure.

Response: As per the suggestion, we have quantified all western blots and represented in the form of group data.

Comment 4: Protein content of all other Sirtuins should be measured in the SIRT6 KO and SIRT6 Tg mice to complement the mRNA data reported.

Response: As per the suggestion, we tested protein levels of all Sirtuins in msSIRT6-KO mice and SIRT6 inhibitor treated mice muscle. Additionally, we have quantified western blots and have represented in the form of group data. Further, we have included mRNA levels of all Sirtuins in msSIRT6-KO mice and SIRT6 inhibitor treated mice muscle. We have also tested activity of other Sirtuins in SIRT6 inhibitor treated mice muscle. Please see Supplementary figure 10C, 10D, Supplementary figure 17A, 17B, Figure 10A. Page 7, 13.

We would like to respectfully point out that SIRT6-Tg mice model has not been used in this study.

Comment 5: Is there a fibre-type shift observed in the SIRT6 KO and Tg mice? COX staining does not achieve this on its own and needs to be complemented with MHC staining.

Response: We thank the reviewer for the thoughtful suggestion to study fibre-type switch in muscle-specific SIRT6 deficient mice muscle. As per the suggestion, we performed MHC staining in msSIRT6-KO mice tibialis anterior muscle at basal level and upon Dex administration. MHC staining results suggest that SIRT6 deficiency in muscle does not affect MHC fiber type proportion at basal level when compared to control mice. Further, we found muscle-specific SIRT6 deficiency abrogates Dex-induced fast to slow fiber switch. Additionally, we have performed MHC staining in SIRT6-inhibitor treated mice. Similar to msSIRT6-KO mice, MHC fiber type proportion in SIRT6-inhibitor treated mice tibialis anterior muscle was comparable to vehicle treated mice muscle. Further, SIRT6-inhibitor treated mice was protected against Dex-induced MHC-fibre type switch. Please see Supplementary figure 13, Supplementary figure 18. Page 8, 13. Please note that we removed COX staining data from the revised manuscript.

Comment 6: Is there a change in habitual energy expenditure or muscle-specific mitochondrial respiration in SIRT6 KO and Tg mice? Both seem important given that the authors are assessing chronic muscle atrophy in which daily activity/mito function will dramatically alter the results.

Response: We thank the reviewer for the thoughtful and constructive comment. As per suggestion, we tested muscle function by performing a treadmill exhaustion test in 19 months old muscle-specific SIRT6 knock-out mice generated using Myogenin-Cre transgenic line. We analyzed four FF myoCre and four F+ myoCre. We recorded the time and distance that the mice run on the treadmill, and oxygen consumption (VO_2), carbon dioxide production (VCO_2), respiratory exchange rate (RQ) and energy expenditure (EE) during the run. Although we observed a trend for the FF myoCre mice to run less than their F+ myoCre littermates, when we analyzed the mice altogether, the difference was not statistically significant, and energy expenditure was not different between genotypes. These data suggest that deletion of SIRT6 is not sufficient to elicit any functional defect in muscle. Please see Figure 6. Page 9.

Comment 7: The authors suggest that the lack of effect of Dex on muscle atrophy in the Quadriceps, Triceps and Biceps (compared to GTN or TA) is due to differing proportion of fast twitch fibres (Figure 1). Please support this with analysis of fibre-type profiles in these muscles. Based on this observation, it is also unclear why the authors did not examine the Soleus muscle?

Response: As per the suggestion, we have measured muscle weight, CSA and fibre types in the Dex treated mice tibialis anterior, gastrocnemius, quadriceps, triceps, biceps, and soleus muscles. MHC staining results revealed Dex induces fast to slow fibre-type switch in all muscles. However, chronic administration of Dex induced atrophy in tibialis anterior, gastrocnemius and soleus muscle. In line with previous reports, we found majority of muscle fibers are of fast type in all muscle except soleus. Dex administration for 24 hrs increased the protein levels of SIRT6 in the tibialis anterior, gastrocnemius and soleus muscles (Fig. 1H and Supplementary Fig. 7A, 7B), the same muscle types which show significant loss of muscle mass upon chronic Dex administration. As per suggestion, we performed all analysis in soleus muscle as well. Please see Figure 1B-1E, 1H, Supplementary figure 1-7. Page 4, 5.

Comment 8: The authors need to demonstrate the specificity of the Atrogin-1, MuRF1 and FOXO3a antibodies they used in their IHC analysis as the majority of the positive staining presented doesn't

appear to actually be in myotubes (rather than in non-differentiated cells). This can be achieved by staining following Atrogin-1, MuRF1 or FOXO3a viral silencing (or peptide competition assays) as they have already been performed for SIRT6.

Response: As suggested, we have tested the specificity of Atrogin-1, MuRF-1/2/3, FoxO1 and FoxO3 antibodies by depletion of Atrogin-1, MuRF-1, FoxO1 or FoxO3 using siRNA against them. We have included representative confocal images and group data showing reduced fluorescence intensity of Atrogin-1, MuRF-1, FoxO1 and FoxO3 in myotubes after their respective knock down. Furthermore, the expression of MuRF2 is high in muscle fibroblasts ¹ and MuRF1/2/3 antibody used in this study detects all isoforms of MuRF. Moreover, previous reports suggest that Atrogin-1, MuRF-1, FoxO1, and FoxO3 express not only in differentiated myotubes but also in undifferentiated myoblasts ²⁻⁴. Therefore, fluorescence signals can be seen in non-differentiated cells as well. However, fluorescence signal from myotubes is only accounted for quantification purposes. Please see Supplementary figure 8, Supplementary figure 14. Page 6, 10.

Comment 9: It would be useful to the reader if the authors also provided the muscle weight data as a % decline, as they have done for body weight (Figure 1B, 3C-G, 6C-G).

Response: As suggested, we have represented change in muscle weight as % decline compared to controls. We have also included absolute muscle weight without normalisation to tibia length in Supplementary figures and in a table format for all the experiments. Please see Figure 1B, Supplementary figure 1B, 2A, 2B, 3A, 3B, 4A, 4B, 5A, 5B, 6A, 6B, Figure 4C, Supplementary figure 12B-12J, Figure 10C, Supplementary figure 17D-17L, and Supplementary table 1, 2, 3.

Comment 10: Specificity of the SIRT6 inhibitor needs to be demonstrated in vivo. Please report changes in SIRT1-7 mRNA, protein and activity for each (Figure 6). The authors should also repeat the AcH3K9 analysis pre and post DEX intervention (i.e. add to the analysis already performed in Figure 6H).

Response: As per suggestion, we tested mRNA and protein levels of all sirtuin isoforms in SIRT6-inhibitor treated mice muscle. We did not observe any significant change in the mRNA or protein levels of the various Sirtuin isoforms except the reduction in SIRT2 mRNA levels in SIRT6 inhibitor treated mice muscle, the reason for which is unclear. We did not investigate this further as the protein levels of SIRT2 were not changed. Furthermore, the acetylation levels of SOD2 or H3K18, targets of SIRT3 ^{5,6} and SIRT7 ^{6,7} respectively, were unaltered in SIRT6 inhibitor treated mice muscle. Additionally, we have previously shown that the acetylation levels of SIRT1 and SIRT2 targets, p53 and α -tubulin respectively also do not change in SIRT6 inhibitor treated mice muscle ⁸. However, we did not test activity of SIRT4 and SIRT5 in SIRT6 inhibitor treated mice muscle since these isoforms show weak deacetylase activity. These data suggest the high specificity of the SIRT6 inhibitor in mice muscle. Please see Figure 10A, Supplementary figure 17A, 17B, Page 13.

As per suggestion, we have included AcH3K9 pre and post Dex administration in SIRT6 inhibitor administered mice muscle. Our result showed that Dex indeed reduced the acetylation level of H3K9 at basal level. However, Dex also decreased the acetylation level of H3K9 in SIRT6 inhibitor administered mice. These results indicate the complex regulation of acetylation status of H3K9 in response to Dex treatment. Notably, Dex has been shown to reduce the acetylation level of H3K9 by controlling its upstream regulators, p300 and HDAC3 ^{9,10}. Therefore, future studies are needed to understand the complex interplay between histone acetylases and deacetylases in Dex treated muscle. Please see Fig. 10G and Supplementary Fig. 19H. Page 13, 14.

Comment 11: The authors generate their SIRT6 models using different Cre lines, but then compare the two models directly. This seems problematic and I think needs to be further highlighted and discussed as a limitation to the data interpretation.

Response: As per suggestion, we have mentioned direct interpretation of data from two msSIRT6-KO mice lines generated using different cre lines as a limitation of our study in discussion section. Please see Page 16, 17.

Comment 12: If the authors are proposing that SIRT6 is driving its biological action via modulation of AKT activity, then it seems pertinent to directly test this through an Akt inhibitor, rather than a PI3K inhibitor. Indeed, LY294002 has been shown to have numerous off target effects (Bain et al PMID: 17850214) that limit interpretation using this compound.

Response: We acknowledge the off-target effects of PI3K inhibitor in our revised manuscript. As per suggestion, we have performed the suggested experiment using AKT inhibitor AKTi-1/2. Our confocal microscopy data analysis suggests AKT inhibition ameliorates the protective effect of SIRT6 depletion against Dex induced reduction in myotube diameter in line with our previous observation using PI3K inhibitor LY294002. Therefore, comparing the results of both the inhibitors together suggest that SIRT6 impacts myotube physiology via modulation of the AKT activity. Please note we have included AKT inhibitor experiment results in main figure and have moved the results of PI3K inhibitor LY294002 experiment in Supplementary figure. Please see Figure 9E, 9F, Supplementary figure 16E, Supplementary figure 16F-16H. Page 12.

Comment 13: The authors suggest that FOXO proteins are regulating some of the biological effects shown, and report data on FOXO1 and FOXO3. It is unclear if they are proposing both to play a role, as in one model they report FOXO3 data (Figure 4) and another FOXO1 data (Figure 5). Can these proteins be used interchangeably? Presumably they are not regulated by universal processes?

Response: We agree with the reviewer that different FoxO isoforms cannot be used interchangeably. The FoxO isoforms are differentially regulated by various post-translational modifications via their upstream modifier enzymes¹¹. FoxO isoforms modulate muscle atrophy by controlling the expression of overlapping target genes, while their effects are distinct under *in vitro* and *in vivo* conditions. FoxO3 regulates Atrogin-1 expression under both *in vitro* and *in vivo* cases, while FoxO1 regulates Atrogin-1 expression under *in vivo* condition but not under *in vitro* condition¹². Yet, distinct set of target genes are also regulated by specific FoxO isoforms¹³. Moreover, deletion of single FoxO isoform is not sufficient to protect against muscle atrophy suggesting overlapping function of FoxO isoforms¹³. AKT mediated phosphorylation of FoxO1, FoxO3 and FoxO4 isoforms protect against Dex induced muscle atrophy in muscle^{14,15}. Therefore, IGF1/PI3K/Akt pathway mediated phosphorylation and inhibition of FoxO is a universally conserved mechanism for coordinated regulation of the activity of FoxO1 and FoxO3 isoforms in muscle for protection against Dex induced muscle atrophy.

To understand the isoform specific role of FoxO in protection against Dex induced muscle atrophy in our study, we performed additional experiments to test: (1) mRNA levels of FoxO3 in Dex administered mice muscle, (2) Protein content of FoxO3 in msSIRT6-KO mice muscle, (3) FoxO1 nucleus localisation in SIRT6 overexpressing myotubes, (4) Protein levels of FoxO1 and FoxO3 in SIRT6 overexpressing myotubes, (5) Protein levels of FoxO1 and FoxO3 in SIRT6 depleted myotubes at basal level and after Dex treatment, (6) Protein levels of FoxO3 in SIRT6-WT or mutant

expressing cells. We found increase in FoxO3 mRNA levels in Dex administered mice muscle similar to FoxO1. Further, we found msSIRT6-KO mice muscle were protected against Dex induced increase in FoxO3 protein levels similar to FoxO1. Next, our confocal microscopy data analysis suggested nuclei localisation of FoxO1 and FoxO3 were significantly increased in SIRT6 overexpressing myotubes. Moreover, proteins levels of FoxO1 and FoxO3 were increased in SIRT6 overexpressing myotubes, while reduced in SIRT6 depleted myotubes, and were further protected against Dex induced increase in FoxO1 and FoxO3 protein levels. In addition, FoxO3 levels increased in ectopically expressing SIRT6-WT cells. These findings suggest IGF1/PI3K/Akt pathway regulates both FoxO1 and FoxO3 isoforms in our model system. Please see 1F, 7D-G, 8C, 9A, 9D, Supplementary figure 15F, 15G, Supplementary figure 16D. Page 5, 10, 11.

Comment 14: Introduction – The authors discuss protein balance and muscle wasting (line 4), however I don't think it is possible to group ageing and the other disorders together as they have as the mechanisms of muscle atrophy are different. Sarcopenia is thought to be driven by a reduction in protein synthesis (not elevations in degradation), whereas the other inflammatory conditions listed are probably a combination of changes in synthesis and degradation.

Response: We thank the reviewer for this insightful comment. We have made suggested changes in the introduction section of revised manuscript. Please see Page 3.

Reviewer #2 comments:

Comment 1: Muscle fibre area is difficult to measure accurately. The approach used by the authors of formalin fixing makes it very difficult to accurately assess muscle fibre area, as the muscle is not pinned at a consistent length. Further to this point, by not pinning the muscles it is very difficult to make an accurate measure of CSA, and it is evident from the WGA images that the fibre sections (at least for some) are cut at an oblique angle. This makes the measurement of fibre area inaccurate. Overall, given the importance of muscle fibre area to the premise of the paper this point should be addressed.

Response: We thank reviewer for the insightful suggestion. As per the suggestion, we repeated CSA analysis by performing WGA staining in muscle sections stored in Poly-Freezer and sectioned by cryostat. We have included representative image with proper cutting. Please see Figure 1C-1E, 4D, 4F, 4G, 10D-10F, and Supplementary figure 1C, 2C, 2D, 3C, 3D, 4C, 4D, 5C, 5D, 6C, 6D, 13A, 18A.

Comment 2: Assessing fibrosis etc. is tricky. More details on the stereological approach is needed. How many sections? What area of the muscle? How much of the muscle cross section was assessed? Was it randomised? An unbiased approach (blinding is mentioned in the Fig legend) is important for accurate measure of histological samples. This applies to the muscle CSA analysis as well.

Response: We thank the reviewer for the suggestion. We have updated method sections on the analysis of fibrosis and CSA. We used formalin fixed muscle tissue for analysing fibrosis. Animals were randomly selected from each group for fibrosis analysis. Fibrosis was analysed manually across the tissue section and multiple non-overlapping frames of the entire tissue were captured from longitudinal sections of the muscle. Three sections per tissue were stained and analysed, while the observer was blinded to the experimental groups. Animals with highest, lowest end and middle of muscle weight were selected for analysing CSA of muscle stored in Poly-Freezer for (1) mice administered with Dex, (2) msSIRT6-KO administered with Dex, while 5 animals were randomly

selected from each group for SIRT6-inhibitor treated mice with/without Dex. Tile scan confocal images were acquired at 10x followed by stitching to generate whole tissue image for the measurement of CSA. One section per tissue was stained with WGA for CSA analysis, while the observer was blinded to the experimental groups. CSA of minimum 200 individual cross-sectional myofibers per muscle or the maximum possible number cross-sectional myofiber were manually analyzed for each muscle using ImageJ software.

Comment 3: The muscle fibres of the mouse are pretty much all "fast" (e.g. PMID: 7932768, 22938020). For example, the quadriceps, TA, triceps, biceps and gastrocnemius are all similar (i.e. predominantly "fast" type 2). The argument that the quadriceps, triceps and biceps did not decrease in weight with glucocorticoid treatment due to fibre type is not accurate. This point should be addressed.

Response: We thank the reviewer for the insightful comment. We agree to reviewer's comment and as per suggestion, we tested the proportion of fiber types by MHC staining in different muscle types. Our analysis shows that tibialis anterior, gastrocnemius, quadriceps, triceps, and biceps muscles are composed majorly of fast type fibers such as MHC IIA, MHC IIB, and MHC IIX fibers, while soleus muscle is comprised of ~ 40% of slow type fiber namely the MHC I as shown previously^{16,17}. Notably, Dex administration induced an increase in slow or MHC I fiber type in all muscle types with concomitant reduction in different fast fiber types in different muscle (MHC IIB and MHC IIX in tibialis anterior and soleus muscle, and MHC IIB in gastrocnemius, and triceps muscles) as shown previously^{18,19}. Furthermore, Dex administration for 24 hrs increased the protein levels of SIRT6 in the tibialis anterior, gastrocnemius and soleus muscles, the same muscle types which show significant loss of muscle mass upon chronic Dex administration. In contrast to these muscle types, the SIRT6 levels were unchanged in quadriceps, triceps, and biceps muscles where the muscle mass also was not significantly changed by Dex administration. Taken together, these results suggest that SIRT6 is specifically upregulated in the muscle types that show significant muscle loss upon Dex administration. Please see Figure 1B-1E, 1H, Supplementary figure 1-7. Page 4,5.

Comment 4: With Dex, was SIRT6 expression differentially effected/increased in the gastrocnemius versus the quadriceps/triceps/biceps? Given muscle weight was differentially effected in these muscles with Dex, per the premise of the study, it would be interesting/important to know if SIRT6 expression was also differentially effected. This should be measured/presented. In fact, this is a very interesting aspect of the study. Why there is a different response in different muscles and how does this fit with the author's findings and premise?

Response: We thank the reviewer for the insightful and constructive comment. As per suggestion, we tested change in muscle weight and SIRT6 levels in the tibialis anterior, gastrocnemius, soleus, quadriceps, triceps, and biceps muscles over the course of the 15 days of Dex administration. Dex administration for 24 hrs increased the protein levels of SIRT6 in the tibialis anterior, gastrocnemius and soleus muscles, the same muscle types which show significant loss of muscle mass upon chronic Dex administration. Notably, the SIRT6 protein levels continued to remain high after 7 and 15 days of Dex administration only in tibialis anterior muscle, but not in other muscle types tested. However, SIRT6 levels reduced significantly after 7 days and 15 days of Dex administration in gastrocnemius muscle. In contrast to these muscle types, the SIRT6 levels were unchanged in quadriceps, triceps, and biceps muscles where the muscle mass also was not significantly changed by Dex administration. However, we did not study change in SIRT6 levels in biceps and soleus muscle after 15 days of Dex administration since tissue size was not sufficient for western analysis in sufficient number for data analysis. Taken together, these results suggest that SIRT6 is specifically upregulated in the muscle

types that show significant muscle loss upon Dex administration and the increase in SIRT6 levels after 24 hr of Dex administration itself reprogramme the proteolytic and protein synthesis machinery leading to change in muscle weight upon chronic Dex administration. Please see Figure 1B-1H, Supplementary figure 1-7. Page 4,5.

In line with these findings, msSIRT6-KO mice and SIRT6 inhibitor treated mice tibialis anterior and gastrocnemius muscles were protected against Dex induced muscle loss. We did not study change in soleus muscle mass after Dex administration in msSIRT6-KO and SIRT6 inhibitor treated mice that remains as a limitation of our study. Please see Figure 4C, Supplementary figure 12B-12J, Figure 10C, Supplementary figure 17D-17L. Page 7, 13.

Comment 5: It is not clear from Figure 1 if SIRT6 increased after or before the atrophy occurred. This point should be made. A nice addition to the paper would be temporal changes in muscle weight and SIRT6 expression over the course of the 15 days of Dex. Regardless, the point above will help address this point.

Response: We thank the reviewer for this thoughtful comment. As per suggestion, we tested changes in muscle weight and levels of SIRT6 over the course of the 15 days of Dex administration. Muscle weight and SIRT6 levels were monitored after 1 day, 7 days and 15 days of Dex administration in different muscle types. We found SIRT6 level increases in tibialis anterior, gastrocnemius, and soleus muscle after 24 hr of Dex treatment that showed concomitant reduction in muscle weight after 15 days of Dex treatment. These results suggest that increase in SIRT6 level at early stage of Dex administration is critical for Dex induced muscle loss after 15 days of Dex administration. Please see Figure 1B, 1H, Supplementary figure 1B, 2A, 2B, 3A, 3B, 7A, 7B. Page 4,5.

Comment 6: Quantification of all western and puromycin blots should be provided throughout the manuscript. Also, for the puromycin blots, did the authors run a ponceau stain (this is the common approach)? That would help verify equal loading, which is critical for such an assay.

Response: As per suggestion, we have performed quantification for all western blots and puromycin assay. We could successfully stain puromycin blots with ponceau for the experiments conducted in *in vivo* samples. However, we were not able to run ponceau staining for the puromycin assay conducted in *in vitro* samples since blots were very old and less protein was loading. Please see Supplementary figure 15M, 19B. Moreover, we have included ponceau stained blot to test equal loading for the blot showing deletion of SIRT6 in different muscles of msSIRT6-KO mice. Please see Supplementary figure 10A.

Comment 7: Figure 3B-G. The number of mice on which body weight was measured is different from the number of mice for which there is muscle weights. It is not clear why this is the case.

Response: Thank you for the suggestion. We have now included all the data point in the figures showing % change in body weight and % change in muscle weight. Please see Figure 4B, 4C, Supplementary figure 12A-12J.

Comment 8: Figure 3B-G. Is the data assessed by 2 way ANOVA? If so, was there a main effect for Dex? Main effects, interaction and post-hoc analysis should be presented. This data is important to the

studies' premise so providing such data will be informative to the reader. The type of analysis should be provided for all figures with similar analysis (that is, 2 x 2 design).

Response: We thank the reviewer for suggestion has improved our data analysis and representation. As per suggestion, we have conducted 2-Way ANOVA analysis with Bonferroni post hoc test for all the experiments of 2 x 2 design. Please see Figure 4B, 4C, Supplementary figure 12B-12J. Further, we have now included main effect, interaction and post-hoc analysis for each experiment with 2-Way ANOVA analysis along with post hoc test in the legend section of the respective figure.

Comment 9: In the msSIRT6KO mice, was SIRT6 comparably knocked out across all muscles?

Response: Thank you for the suggestion. As per suggestion, we have tested SIRT6 level in different muscle from SIRT6-fl/fl and msSIRT6-KO mice muscle. Our western blot data suggested SIRT6 was comparably deleted in various muscles of msSIRT6-KO mice. Please see Figure 4A (right panel). Moreover, we have included ponceau stained blot to test equal loading for the blot showing deletion of SIRT6 in different muscles of msSIRT6-KO mice. Please see Supplementary figure 10A. Page 6.

Comment 10: For Dex treatment of mice, rather than % change in body weight it would be helpful to show the pre/post body weight for each mouse (with lines joining them). This could be a Suppl figure, if needed.

Response: As per suggestion, we have included line graph showing change in body weight with time upon Dex treatment for each mouse in different mice models used in our study. Please see Supplementary figure 1A, 12A, 17C.

Comment 11: Are the analyses performed in male or female mice? If one sex, are the findings of primary measures consistent in the other sex? Please update Methods with details.

Response: The msSIRT6-KO mice experiments were conducted in both male and female mice and so was data analysis. Further, comparable effects of Dex and SIRT6 deficiency on muscle phenotype was observed in both sexes. We have updated method section accordingly. Please see Page 26.

Comment 12: The authors measured muscle weight on 7-10 mice, yet the fibre area is on only 4 mice. How did the authors select which mice to run CSA measures on? Was this randomized? CSA data for all mice on which muscle weight was analysed should be presented. At the every least, a distribution of mice that represents the high, middle and low end of muscle weight would be helpful.

Response: Mice were randomly selected for CSA analysis from each group in the initial submission of the manuscript. We could not perform CSA analysis for all animals on which muscle weight was analysed due to resource and funding limitation. Therefore, as per the suggestion, we have selected animals with highest, middle, and lowest end of muscle weight for re-analysing CSA of tibialis anterior muscle stored in PolyFreeze. Analysis of CSA suggested msSIRT6-KO mice were protected against Dex induced muscle atrophy in line with analysis presented during the initial submission. Please see Figure 4D, 4F, 4G and Supplementary figure 13A.

Comment 13: The breeding strategy for the msSIRT6KO mice is unclear. Are the fl/fl mice, littermate controls? Or, were 2 separate lines of mice bred to provide the Cre+ and Cre- mice? Please provide details.

Response: As per suggestion, we have included breeding strategy for generating msSIRT6-KO in method section. SIRT6-fl/fl mice and Acta-Cre mice were obtained from Jackson Laboratory. Muscle-specific SIRT6 knockout mice were generated by crossing SIRT6-fl/fl mice with Acta-Cre,SIRT6-fl/+ mice to obtain Acta-Cre,SIRT6-fl/fl mice. The SIRT6-fl/fl littermates were used as controls for experiment. For the second line, C57BL/6-SIRT6 conditional mice (SIRT6fl/fl) that were previously generated²⁰ were crossed with C57BL/6 mice transgenic for the Cre recombinase under the control of the muscle-specific myogenin promoter (gift from P. Puigserver), Sirt6^{F/+} to achieve deletion of SIRT6 specifically in muscle. Sirt6^{F/F} Myogenin-cre mice were compared with the heterozygous Sirt6^{F/+} Myogenin cre mice that may or may not be littermates. Please see Page 26, 27.

Comment 14: Figure 3L. What are the results of the 2 way ANOVA analysis for this figure? Are there differences?

Response: As per suggestion, we performed 2-Way ANOVA analysis with Bonferroni post hoc test for Figures 4H and 4I in revised version of manuscript. We increased number of animals for these experiments and re-analysed the data. Our 2-Way ANOVA analysis with Bonferroni post hoc test suggested expression of Atrogin-1 was significantly low in msSIRT6-KO mice muscle at basal level and msSIRT6-KO mice were resistant against Dex induced increased expression of Atrogin-1. However, msSIRT6-KO mice muscle showed trend towards reduction in MuRF-1 mRNA level at basal level but was not significant. In line with Atrogin-1 expression, msSIRT6-KO were resistant to Dex induced increased expression of MuRF-1. 2-Way ANOVA analysis with Bonferroni post hoc test suggested significant interaction between Dex and genotype and the effect of both factors were extremely significant and acted as main effects. We have updated 2-Way ANOVA analysis results in figure legend. Please see Figure 4H, 4I.

Comment 15: Related to this statement: However, we also observe myofiber enlargement at basal conditions in these animals, further validating a role for SIRT6 in modulating fiber size (Fig. S3D-E). This data is a combination of two different muscles? This is not an appropriate representation of the data. It is also not appropriate to present individual fibre data. Please separate the two muscle types and just present the average fiber area.

Response: Indeed, the data presented in that figure (current Fig. 5F,G), includes measurements made on both the soleus and gastrocnemius muscle. At that time, given the limited number of tissue we had available, we decided to combined the measurements of all fibers in order to increase statistical power. Unfortunately, due to COVID we have lost the strain, and we do not have the possibility to perform these experiments in a different manner. But we will be happy to tone down the conclusions. Of note, these experiments are presented only as confirmatory for the other muscle-specific Cre deleted mice (ACTA1-Cre), where the analysis was done on individual muscles and the results were the same.

Comment 16: FoxO transcriptional activity is reduced ~50% in SIRT6-KD cells. Thus, there is still 50% activity. Do the authors think that there is a 'tipping' point below which FoxO activity is sufficiently reduced such that it is protective? Physiologically speaking, does this seem reasonable?

Response: We thank the reviewer for the comment. We acknowledge that FoxO activity was reduced ~50% in SIRT6 depleted myotube. The change in FoxO activity was seen using SIRT6 specific siRNA that reduces SIRT6 levels in myotubes by ~50% as shown in Figure 3A. Moreover, the conclusion drawn from knockdown experiments are based on the myotube phenotype observed by

using ~50% efficient siRNA. As it stands, we cannot judge the effect of more efficient SIRT6 siRNA on FoxO activity. Therefore, although there may be a putative 'tipping' point for reduction in FoxO activity, with the data we have, it is impossible for us to make that conclusion.

Comment 17: Please provide quantification of the blots in Fig 5. How long was Dex administered for in this data (please state in text of Fig Legend)?

Response: As per suggestion, we have included quantification for figure 9A, 9C, 9D in supplementary figures. Dex was administered for 15 days at the dose of 10 mg/kg/day. We have updated the Figure legend as per suggestion. Please see Figure 9, Supplementary figure 15, 16A-16D.

Comment 18: Also, in Fig 5A, SIRT6 protein did not increase (looks like it decreases?) with Dex in fl/fl mice. This does not fit with the model proposed by the authors, nor the data in Figure 1. Some comment in the text is needed.

Response: We agree with the reviewer regarding reduction in SIRT6 level in the figure 9A in the revised manuscript. The representative western blot shows SIRT6 level in mice muscle after 15 days of Dex administration in gastrocnemius muscle. We performed a time point experiment to test change in SIRT6 level after 24 hrs, 7 days, and 15 days of Dex treatment. Our data suggest that SIRT6 level increases after 24 hrs but reduces significantly after 7 days and 15 days of Dex treatment in gastrocnemius muscle. Please see Supplementary figure 7A. Moreover, we have included group data quantification for figure 9A of revised manuscript. Please see Figure 9A, Supplementary figure 15N. Please see Page 5, 11.

Comment 19: Please provide quantification of Figs 6A and 6H. What is the n for these experiments?

Response: We have included group data showing quantification. We have updated the number of animals for these experiments in figure legend. Please see Figure 10A, 10G, Supplementary figure 19.

Comment 20: Similar to above, for Fig 6B-G, please provide the results of the 2 way ANOVA table analysis in the figure.

Response: As per suggestion, we have included 2-way ANOVA analysis in the figure legend. Please see Figure 10B, 10C, Supplementary figure 17C-17L.

Comment 21: The muscle-specific SIRT6 KO mice is a nice model. However, the authors used a germline Cre approach. The limitations of this approach should also be addressed/acknowledged.

Response: We thank the reviewer for suggestion to include the limitation of the germline-cre approach. As per suggestion, we have included the limitation of this approach in discussion section. Please see page 16, 17.

Comment 22: The relevance of when Cre turns on during development in MCK vs HSA Cre mice, as is discussed in the Discussion, is unclear. Why does it matter when it turns on in terms of effects on measures that are made?

Response: As per the suggestion, we have removed the above-mentioned statement from the revised manuscript.

Comment 23: The 3rd paragraph of the Discussion does not seem relevant to the paper. Or, could be moved toward the end of the Discussion if it is felt that it adds value.

Response: As per suggestion, we have rearranged discussion section.

Comment 24: Do the authors think that their mechanism works through Akt1? Some discussion on Akt isoforms in muscle is warranted.

Response: We thank reviewer for the suggestion. We have not performed any experiment to identify the AKT isoform responsible for protection of muscle-specific SIRT6-deficient mice muscle against Dex-induced muscle atrophy. However, we do speculate that proposed mechanism works through combined action of AKT1/AKT2. As per suggestion, we have discussed about the role of AKT isoforms in the regulation of muscle homeostasis. Please see Page 15.

Comment 25: Given SIRT6 KO in muscle has previously been shown to effect AMPK activity, do the authors find any effect on AMPK in their model (e.g. pAMPK or pACC)? This would be a nice inclusion to the paper.

Response: As per suggestion, we tested activating phosphorylation of AMPK and total AMPK in msSIRT6-KO. We found reduced p-AMPK in msSSIRT6-KO mice muscle and further msSIRT6-KO mice were resistant against Dex-induced increase in p-AMPK. Our findings are in line with previous reports suggesting inhibitory action of AMPK on muscle protein synthesis by negatively regulating mTOR signaling²¹. AMPK also promotes transcriptional activity of FoxO and increases atrogenes, Atrogin-1 and MuRF-1 expression²¹. However, we did not perform any rescue experiment using AMPK inhibitor in msSIRT6-KO mice or SIRT6 depleted myotubes. Therefore, we speculate AMPK signaling might be involved in protection of muscle-specific SIRT6 deficient mice muscle against Dex-induced atrophy, although more experiments will be needed in the future to fully understand the role of AMPK. Please see Figure 9A, Supplementary figure 15J, 15K. Please see Page 11.

Comment 26: Most of the analysis has an n=3 or 4 mice. This is very low for mouse studies, especially of muscle biology. What is the rationale for this?

Response: We used low number of animals for some of experiments during initial submission of manuscript due to limitations in funding and resources. As per suggestion, we have repeated analysis with more animals for majority of experiments where number of animals were low. Moreover, we have updated n for each group in majority of experiments in figure legend.

Comment 27: Absolute (i.e. not corrected for tibia length) muscle weights should be presented in a table. This could go in the Suppl Files. Given that the authors find no effects on body weight then the absolute data will be nice to see. As an aside, was tibia length different between mouse models?

Response: We thank the reviewer for the suggestion. As per suggestion, we evaluated tibia length of mice and have included a table showing tibia length for all mice models. Further, we did not find any change in the tibia length of mice for any of the mice models since muscle-specific SIRT6 deficiency does not affect growth of mice. Besides, we have represented absolute muscle weight as Scatterplot as well as in table format in Supplementary figures and Supplementary table respectively. Please see

Supplementary figure 1B, 2B, 3B, 4B, 5B, 6B, 12B, 12D, 12F, 12H, 12J, 17D, 17F, 17H, 17J, 17JL and Supplementary table 1, 2, and 3.

Comment 28: Was total muscle protein concentration differentially effected in Dex and SIRT6 KO mice? This would be a nice addition related to atrophy.

Response: We thank the reviewer for thoughtful comment. As per comment, we have tested protein concentration in msSIRT6-KO mice at basal level and upon Dex administration by using Bradford assay. Our result showed that soluble protein concentration in msSIRT6-KO mice were comparable to controls at basal level and also upon Dex administration. However, Dex treated control mice showed a trend towards decrease in soluble protein concentration, although the changes were not statistically significant. These results indicate that Dex-induced phenotypic changes in control mice are independent of soluble protein levels in muscle. Please see Supplementary figure 12K. Please see Page 8.

Comment 29: Was protein synthesis effected in msSIRT6 KO mice? Such analysis would tie in nicely with the differences in fibre area etc.

Response: We thank the reviewer for the thoughtful comment. We conducted puromycin incorporation assay in msSIRT6-KO mice treated with Dex. Our puromycin incorporation assay analysis suggested rate of protein synthesis was increased in msSIRT6-KO mice muscle at basal level. Further, msSIRT6-KO mice were resistant against Dex-induced reduction in rate of protein synthesis. Please see Figure 9A, Supplementary figure 15L, 15M. Please see Page 11.

Minor comments:

Comment 30: For the gastrocnemius weights, was this the gastrocnemius complex, or was the plantaris muscle dissected out? Please provide details.

Response: We dissected out gastrocnemius muscle complex and this information is updated in the method section. Please see Page 26.

Comment 31: In Figure 1J, the SIRT6 blot looks like a spliced blot. If this is the case, please mark the blot as such.

Response: We have attached the marked blots used during initial submission of the manuscript. Please note that we used middle 4 lanes during initial submission as shown below. However, we repeated the experiment and replaced panel with better representative blots. We used unspliced blot in revised manuscript. Further, we quantified the blots and included group data in the revised manuscript. Please see Figure 1M in the revised manuscript.

Comment 32: The post-hoc test used for ANOVA analyses is not mentioned.

Response: We have updated post-hoc test used for ANOVA analysis in Figure legend section.

Comment 33: It would help the reader if the y-axis (or above the figure) had more information about the measure being made. Otherwise, one needs to keep referring to the text or Fig Legend.

Response: We have updated figures as per the suggestion.

Reviewer #3 comments:

Comment 1: Results, page 4: the authors claim that they have established an experimental model of muscle atrophy based on DEX administration. However, this model is widely used and very well characterized since long ago. In this regard, the only novelty is the up-regulation of SIRT6 expression. I would encourage the authors to clearly indicate in the text that the data reported in Figure 1, panels A-H, are just confirmatory;

Response: We thank the reviewer for the thoughtful suggestion. Indeed, Dex-induced muscle atrophy is a well-established model and as per the suggestion we have included the statement that our data are just confirmatory for a well characterised model of muscle atrophy. Further, we have included the statement that we identified change in SIRT6 levels using this atrophy model. Please see Page 5.

Comment 2: Results page 4: the authors state that they validated the in vivo data with results obtained in C2C12 cultures exposed to DEX. How can cell cultures be used to validate an in vivo study?

Response: We thank the reviewer for the constructive suggestion. As per the suggestion, we have corrected the text in revised manuscript. Please see Page 4, 5.

Comment 3: the definition 'myotube atrophy' is not appropriate, better saying 'reduction/decrease in size', 'smaller than', etc.;

Response: We thank the reviewer for the insightful suggestion. Further, we have changed the definition 'myotube atrophy' as per the suggestion in revised manuscript.

Comment 4: SIRT6 KO mice appear comparable to controls, in particular no changes in body weight and muscle mass are reported. However, muscle cross sectional area (CSA) is increased while the expression of molecules involved in the proteolytic machinery is reduced, suggesting that the lack of SIRT6 affects muscle homeostasis also in the absence of a catabolic stimulus (in this case, DEX). How do the authors explain that such effects are not reflected on muscle mass?

Response: We thank the reviewer for the insightful comment. We found change in tibialis anterior and gastrocnemius muscle weight, but weight of quadriceps, triceps and biceps of Dex treated msSIRT6-KO mice remained similar to control mice. Furthermore, body weight of muscle-specific SIRT6 knock-out mice were comparable to control mice. The tibialis anterior and gastrocnemius muscle weight account for a very small fraction of the total weight of an animal. Therefore, we speculate that the change in only tibialis anterior and gastrocnemius weight will not significantly affect the total weight of the mice.

To better understand the reason behind the discrepancy observed between molecular and phenotypic changes in msSIRT6-KO mice tibialis anterior muscle, we evaluated proportion of fiber types by MHC staining. MHC fiber types in msSIRT6-KO mice were comparable to controls at basal level.

Interestingly, msSIRT6-KO mice tibialis anterior muscle was protected against Dex induced fast to slow fiber type switch. Please see Supplementary figure 13. Further, COX staining revealed larger fibre size in fast fibers in both ACTA cre and Myogenin cre-generated msSIRT6-KO mice line (Data not shown). Therefore, we speculate possible reasons as follows: (1) Increase in CSA might be fibre type specific. However, we did not perform MHC staining and CSA together to draw any conclusion in this regard. (2) Increase in fibre number in msSIRT6-KO mice tibialis anterior muscle. Moreover, we measured CSA and MHC staining in only tibialis anterior muscle of msSIRT6-KO mice, a limitation of this study.

Though muscle mass accounts for a major portion in the body weight, we did not find any significant effect of muscle-specific deletion of SIRT6 on body weight. Therefore, we speculate that SIRT6 might not have a generalised action on all muscle types in the body, and further studies are needed to understand the generalised function of SIRT6 in the muscle.

Comment 5: strictly related to the previous point, all the in vivo stuff lacks very important information on muscle function. While it is known that DEX impinges also on muscle contractile activity, nothing is demonstrated in the present study when SIRT6 is manipulated (both deleted and overexpressed);

Response: We thank the reviewer for the thoughtful comment. As per suggestion, we tested muscle function by performing a treadmill exhaustion test in 19 months old muscle-specific SIRT6 knock-out mice generated using Myogenin-Cre transgenic line. We analyzed four FF myoCre and four F+ myoCre. We recorded the time and distance that the mice run on the treadmill, and oxygen consumption (VO_2), carbon dioxide production (VCO_2), respiratory exchange rate (RQ) and energy expenditure (EE) during the run. Although we observed a trend for the FF myoCre mice to run less than their F+ myoCre littermates, when we analyzed the mice altogether, the difference was not statistically significant, and energy expenditure was not different between genotypes. These data suggest that deletion of SIRT6 is not sufficient to elicit any functional defect in muscle. Please see Figure 6. Page 9.

Comment 6: the authors justify the striking discrepancy among their data and previously published results (ref 34 in the paper), discussing that while these latter were obtained using a total body SIRT6 KO, their model relies on a muscle-specific deletion. This is totally reasonable, however, the authors obtain protection against muscle atrophy also using a pharmacological SIRT6 inhibitor, that is practically mimicking the total body KO. How do they explain such a different pattern?

Response: We thank the reviewer for giving us the opportunity to explain the reason behind the discrepancy in phenotype observed between whole-body SIRT6 knock-out mice and muscle-specific SIRT6 deficient mice. Whole-body SIRT6 knock-out mice develop severe muscle atrophy as reported previously²², and also observed in our laboratory (Data not published). The whole-body SIRT6-KO mice develop accelerated aging like phenotype. The degenerative changes observed in whole-body SIRT6-KO mice are not limited to muscle but seen in many organs. The degenerative changes in the whole-body SIRT6-KO mice are associated with severe metabolic defects including severe hypoglycemia. Therefore, the observed degenerative changes in muscle might not be the direct effect of SIRT6 function in muscle but could be an indirect effect of systemic complications such as severe hypoglycemia or distant organs-dysfunction. Therefore, many tissue-specific SIRT6 deleted mice lines have been generated to study tissue-specific function of SIRT6. Moreover, many tissue-specific

SIRT6 deficient mice lines do not develop aging-like phenotype and severe complications as seen in whole-body SIRT6-KO mice similar to our mice model²³⁻²⁶.

Whole body SIRT6 knockout mice show absence of SIRT6 mRNA and protein in all cell types. However, SIRT6 inhibitor specifically inhibits the catalytic activity of SIRT6 without targeting mRNA and protein levels of SIRT6 that will have restricted impact on body when compared to whole body deletion of SIRT6. Emerging roles of SIRT6 suggest its catalytic independent functions in different cell types^{27,28}. The idea of using a SIRT6 inhibitor in this study fits well with deacetylase activity dependent function of SIRT6 in protection against Dex-induced muscle loss. Therefore, it will not affect those SIRT6 functions which are independent of SIRT6 catalytic activity. More importantly, the effect of pharmacological inhibition as used in this study is limited to a shorter period of time in contrast to the generalized long-term impact of whole body SIRT6 deficiency on body. Further, we acknowledge the off-target effect of SIRT6 inhibitor in mice. We have included these points in discussion. Please see Page 16.

Comment 7: Figure 1, panel B: please check point distribution for the Triceps muscle;

Response: As per suggestion, we have re-made the figure showing clearer distribution of all data points. Please see Supplementary figure 5A, 5B.

Comment 8: immunofluorescence experiments should be confirmed by western blotting;

Response: We thank the reviewer for the suggestion. We have performed western blotting to test FoxO1 and Foxo3 protein content in SIRT6 overexpressing and SIRT6-depleted primary myotube treated with Dex. Further, we have included quantification showing group data. However, we were not able to perform western blotting for Atrogin-1 and MurF-1 since the antibodies against Atrogin-1 and MurF-1/2/3 used in this study did not work for western blotting technique. Please see Figure 7G, 8C. Please see Page 10.

Comment 9: DEX effect on protein synthesis is very clear in Figure 1H but not in Figures 2K and 6H;

Response: We thank the reviewer for the suggestion. We have updated western blots for puromycin assay with the better representation of the group data. Furthermore, we have included group data in the form of scatterplot showing quantification in the revised manuscript. Please see Figure 1K, 2F, 3F, 9A, 10G, Supplementary figure 15L, Supplementary figure 19A.

Comment 10: the effect of SIRT6 inhibitor should also be evaluated on myofiber CSA, the more so since SIRT6 genetic ablation results in basal CSA enlargement;

Response: We thank the reviewer for the thoughtful suggestion. As per suggestion, we evaluated CSA in SIRT6 inhibitor treated mice tibialis anterior muscle at basal level and after Dex administration. In line with msSIRT6-KO results, SIRT6 inhibitor treated mice tibialis anterior muscle showed increase in CSA at basal level and were resistant against Dex induced reduction in myofiber CSA. Please see Figure 10D-10F, Supplementary figure 18A. Please see Page 13.

Comment 11: Figure 5A, 5B and 5C, empty vector should be shown as well.

Response: Please note that Figure 9A, 9B and 9C in the revised manuscript are representing *in vivo* experiments. We used SIRT6-fl/fl mice as controls for these experiments. Empty vector was used as control for *in vitro* experiments as shown in Figure 9D of revised manuscript and are not needed for *in vivo* experiments.

REFERENCES

- 1 Silvestre, J. G. *et al.* The E3 ligase MuRF2 plays a key role in the functional capacity of skeletal muscle fibroblasts. *Braz J Med Biol Res* **52**, e8551, doi:10.1590/1414-431X20198551 (2019).
- 2 Lokireddy, S. *et al.* Identification of atrogin-1-targeted proteins during the myostatin-induced skeletal muscle wasting. *Am J Physiol Cell Physiol* **303**, C512-529, doi:10.1152/ajpcell.00402.2011 (2012).
- 3 Moriscot, A. *et al.* MuRF1 and MuRF2 are key players in skeletal muscle regeneration involving myogenic deficit and deregulation of the chromatin remodeling complex. *JCSM Rapid Communications* **2** (2019).
- 4 Hu, P., Geles, K. G., Paik, J. H., DePinho, R. A. & Tjian, R. Codependent activators direct myoblast-specific MyoD transcription. *Dev Cell* **15**, 534-546, doi:10.1016/j.devcel.2008.08.018 (2008).
- 5 Chen, Y. *et al.* Tumour suppressor SIRT3 deacetylates and activates manganese superoxide dismutase to scavenge ROS. *EMBO Rep* **12**, 534-541, doi:10.1038/embor.2011.65 (2011).
- 6 Schiedel, M., Robaa, D., Rumpf, T., Sippl, W. & Jung, M. The Current State of NAD(+) - Dependent Histone Deacetylases (Sirtuins) as Novel Therapeutic Targets. *Med Res Rev* **38**, 147-200, doi:10.1002/med.21436 (2018).
- 7 Barber, M. F. *et al.* SIRT7 links H3K18 deacetylation to maintenance of oncogenic transformation. *Nature* **487**, 114-118, doi:10.1038/nature11043 (2012).
- 8 Sociali, G. *et al.* Pharmacological Sirt6 inhibition improves glucose tolerance in a type 2 diabetes mouse model. *FASEB J* **31**, 3138-3149, doi:10.1096/fj.201601294R (2017).
- 9 Yang, H., Menconi, M. J., Wei, W., Petkova, V. & Hasselgren, P. O. Dexamethasone upregulates the expression of the nuclear cofactor p300 and its interaction with C/EBPbeta in cultured myotubes. *J Cell Biochem* **94**, 1058-1067, doi:10.1002/jcb.20371 (2005).
- 10 Huang, W., Zhou, J., Zhang, G., Zhang, Y. & Wang, H. Decreased H3K9 acetylation level of LXRA mediated dexamethasone-induced placental cholesterol transport dysfunction. *Biochim Biophys Acta Mol Cell Biol Lipids* **1864**, 158524, doi:10.1016/j.bbalip.2019.158524 (2019).
- 11 Wang, Z., Yu, T. & Huang, P. Post-translational modifications of FOXO family proteins (Review). *Mol Med Rep* **14**, 4931-4941, doi:10.3892/mmr.2016.5867 (2016).
- 12 Stitt, T. N. *et al.* The IGF-1/PI3K/Akt pathway prevents expression of muscle atrophy-induced ubiquitin ligases by inhibiting FOXO transcription factors. *Mol Cell* **14**, 395-403, doi:10.1016/s1097-2765(04)00211-4 (2004).
- 13 Milan, G. *et al.* Regulation of autophagy and the ubiquitin-proteasome system by the FoxO transcriptional network during muscle atrophy. *Nat Commun* **6**, 6670, doi:10.1038/ncomms7670 (2015).
- 14 Tran, H., Brunet, A., Griffith, E. C. & Greenberg, M. E. The many forks in FOXO's road. *Sci STKE* **2003**, RE5, doi:10.1126/stke.2003.172.re5 (2003).
- 15 Sandri, M. *et al.* Foxo transcription factors induce the atrophy-related ubiquitin ligase atrogin-1 and cause skeletal muscle atrophy. *Cell* **117**, 399-412, doi:10.1016/s0092-8674(04)00400-3 (2004).
- 16 Mathewson, M. A., Chapman, M. A., Hentzen, E. R., Friden, J. & Lieber, R. L. Anatomical, architectural, and biochemical diversity of the murine forelimb muscles. *J Anat* **221**, 443-451, doi:10.1111/j.1469-7580.2012.01559.x (2012).

- 17 Timson, B. F., Bowlin, B. K., Dudenhoeffer, G. A. & George, J. B. Fiber number, area, and composition of mouse soleus muscle following enlargement. *J Appl Physiol* (1985) **58**, 619-624, doi:10.1152/jappl.1985.58.2.619 (1985).
- 18 Polla, B., Bottinelli, R., Sandoli, D., Sardi, C. & Reggiani, C. Cortisone-induced changes in myosin heavy chain distribution in respiratory and hindlimb muscles. *Acta Physiol Scand* **151**, 353-361, doi:10.1111/j.1748-1716.1994.tb09754.x (1994).
- 19 Pellegrino, M. A. *et al.* Clenbuterol antagonizes glucocorticoid-induced atrophy and fibre type transformation in mice. *Exp Physiol* **89**, 89-100, doi:10.1113/expphysiol.2003.002609 (2004).
- 20 Sebastian, C. *et al.* The histone deacetylase SIRT6 is a tumor suppressor that controls cancer metabolism. *Cell* **151**, 1185-1199, doi:10.1016/j.cell.2012.10.047 (2012).
- 21 Thomson, D. M. The Role of AMPK in the Regulation of Skeletal Muscle Size, Hypertrophy, and Regeneration. *Int J Mol Sci* **19**, doi:10.3390/ijms19103125 (2018).
- 22 Samant, S. A., Kanwal, A., Pillai, V. B., Bao, R. & Gupta, M. P. The histone deacetylase SIRT6 blocks myostatin expression and development of muscle atrophy. *Sci Rep* **7**, 11877, doi:10.1038/s41598-017-10838-5 (2017).
- 23 Cui, X. *et al.* SIRT6 regulates metabolic homeostasis in skeletal muscle through activation of AMPK. *Am J Physiol Endocrinol Metab* **313**, E493-E505, doi:10.1152/ajpendo.00122.2017 (2017).
- 24 Kim, H. S. *et al.* Hepatic-specific disruption of SIRT6 in mice results in fatty liver formation due to enhanced glycolysis and triglyceride synthesis. *Cell Metab* **12**, 224-236, doi:10.1016/j.cmet.2010.06.009 (2010).
- 25 Guo, J. *et al.* Endothelial SIRT6 Is Vital to Prevent Hypertension and Associated Cardiorenal Injury Through Targeting Nkx3.2-GATA5 Signaling. *Circ Res* **124**, 1448-1461, doi:10.1161/CIRCRESAHA.118.314032 (2019).
- 26 Sundaresan, N. R. *et al.* The sirtuin SIRT6 blocks IGF-Akt signaling and development of cardiac hypertrophy by targeting c-Jun. *Nat Med* **18**, 1643-1650, doi:10.1038/nm.2961 (2012).
- 27 Ravi, V. *et al.* SIRT6 transcriptionally regulates global protein synthesis through transcription factor Sp1 independent of its deacetylase activity. *Nucleic Acids Res* **47**, 9115-9131, doi:10.1093/nar/gkz648 (2019).
- 28 Peng, L. *et al.* Deacetylase-independent function of SIRT6 couples GATA4 transcription factor and epigenetic activation against cardiomyocyte apoptosis. *Nucleic Acids Res* **48**, 4992-5005, doi:10.1093/nar/gkaa214 (2020).

Reviewers' Comments:

Reviewer #1:

Remarks to the Author:

I thank the authors for a detailed response to the comments and queries I raised following the first round of review. The authors should be commended for the additional work they have performed which I believe has improved the clarity of the data presented. Congratulations on an interesting study.

Reviewer #2:

Remarks to the Author:

The authors have made commendable and substantial additions and changes to the manuscript and have been receptive to the Reviewers comments. This includes the addition of a substantial analysis. However, I have some additional points to address.

Muscle atrophy can be impacted by physical activity. While the authors undertook the treadmill exercise for a marker activity it does not give insight into cage (i.e. voluntary) activity. This would be the main parameter that could impact muscle atrophy. Is this different? Previous studies show important effects of SIRT6 KO on cage activity (PMID: 28765271). The question here is whether changes in cage activity are part of the reason for the effect(s) on muscle atrophy.

The AMPK results are interesting. Don't they potentially suggest that the regulatory effect(s) of SIRT6 are 'simply' a secondary effect of changes in AMPK activity? One wonders if AMPK inhibition prevents the effects of SIRT6 modulation on measured parameters. While such an experiment in mice would be ideal it is understood that this is a considerable amount of work. But a cell culture experiment could answer this. Or, perhaps in the SIRT6 inhibitor experiments AMPK activity could also be measured.

It is important to note that in reference 51 (on Akt 1 and 2 KO mice) that AMPK activity was also impacted. As such, the discussion on page 15 related to Akt1 and Akt2 should also include discussion about AMPK, such that "SIRT6 controls the activity of PI3K/AKT signaling while protecting against Dex induced muscle mass loss through the combined action of AKT1/AKT2" and potentially AMPK.

Although not critical for a resubmission, a nice addition to the paper would be a more robust measure of AMPK activity, be it pACC or an activity assay.

The title and abstract is misleading as it infers (at least to the Reviewer/reader) that SIRT6 is "directly" modulating glucocorticoid-induced skeletal muscle atrophy. It is clear that these effects are secondary to a regulatory axis that includes AKT1/2 and potentially AMPK. The title and abstract, and conclusions in general, should more accurately reflect the data and mechanism of action.

Given that hierarchically speaking, SIRT6 has many pathways that it regulates, in many different tissues, to suggest targeting SIRT6 as a potential strategy for treating muscle wasting associated with stress and disease conditions seems premature and potentially problematic. How would you do so specifically in skeletal muscle so as to avert off target effects/consequences in other tissues? Also, why not modulate Akt (or AMPK), given that it works through such a mechanism? The discussion of modulating SIRT6 as a means of treating muscle wasting should be balanced by potential issues with such an approach, given the broad contexts of SIRT6 action.

Related to the ponceau stains for the puromycin blots. Please provide quantification of the ponceau stains. Also, it is unclear if the SUnSET analysis corrected for the ponceau stained membranes? For those SUnSET membranes that a ponceau stain exists, the puromycin should be corrected for ponceau. It is also unclear if the SUnSET is corrected for GAPDH/loading control? Please state in Methods.

While the unexpected recombination events is a concern, a major concern with the germline approach, which is not stated, is secondary (or tertiary etc.) changes/adaptations that occur in the muscle as a result of long-term loss of SIRT6. Reference 51 is a perfect example of such a problem in that loss of Akt1/2 resulted in AMPK activation. This can result in misinterpretation of the role of the protein that is, for example, knocked out. This point about secondary etc. changes should also be incorporated into the relevant paragraph and tied back to the author's results.

The manuscript is difficult to read in places. A robust review for syntax and grammar is recommended. One example is the last sentence of paragraph 2 of the Discussion.

Reviewer #3:

Remarks to the Author:

The present version of the study by Mishra and collaborators is markedly improved in comparison to the previous one. The authors satisfactorily addressed all the concerns raised in the first revision round. However, there is still one single issue that needs to be clarified, referred to author comment to point n. 4. In this regard, it is unclear if the hypothesis provided by the authors to discuss the differences occurring between molecular and phenotype discrepancies are reported somewhere in the paper. If not, they should...

Reviewer #1

Comment 1: I thank the authors for a detailed response to the comments and queries I raised following the first round of review. The authors should be commended for the additional work they have performed which I believe has improved the clarity of the data presented. Congratulations on an interesting study.

Response: We thank the reviewer for appreciating our revised manuscript.

Reviewer #2

The authors have made commendable and substantial additions and changes to the manuscript and have been receptive to the Reviewers comments. This includes the addition of a substantial analysis. However, I have some additional points to address.

Comment 1: Muscle atrophy can be impacted by physical activity. While the authors undertook the treadmill exercise for a marker activity it does not give insight into cage (i.e. voluntary) activity. This would be the main parameter that could impact muscle atrophy. Is this different? Previous studies show important effects of SIRT6 KO on cage activity (PMID: 28765271). The question here is whether changes in cage activity are part of the reason for the effect(s) on muscle atrophy.

Response: We thank the reviewer for the suggestion. We agree that treadmill activity does not give insight into cage (i.e. voluntary) activity. However, we would like to bring to your attention that we performed metabolic cage experiments in SIRT6 deficient mice. Our results suggested that muscle-specific deletion of SIRT6 does not significantly change oxygen consumption, carbon dioxide production, respiratory exchange rate, and energy expenditure. Unfortunately, we were not able to present these findings in fine detail in the previously submitted version of the manuscript. We have revised the manuscript for better clarity. Please see page 9. The previous publication (PMID: 28765271) showed a significant effect of SIRT6 deficiency on cage activity. These results contrast with our findings which might be due to the difference in Cre promoter used in the studies. While the authors

generated muscle-specific SIRT6 deficient animal using the Creatine Cre promoter, our model was generated using myogenin cre. We have now included a brief comment about this in the discussion section. Please see Page 16.

Comment 2: The AMPK results are interesting. Don't they potentially suggest that the regulatory effect(s) of SIRT6 are 'simply' a secondary effect of changes in AMPK activity? One wonders in AMPK inhibition prevents the effects of SIRT6 modulation on measured parameters. While such an experiment in mice would be ideal it is understood that this is a considerable amount of work. But a cell culture experiment could answer this. Or, perhaps in the SIRT6 inhibitor experiments AMPK activity could also be measured.

Response: We thank the reviewer for the insightful comment. We first analysed the change in AMPK activity by testing ACC phosphorylation, an AMPK downstream target in Veh or Dex treated msSIRT6-KO. While Dex-treated control mice showed increased ACC phosphorylation, msSIRT6-KO mice were resistant against Dex-induced increase in ACC phosphorylation. Furthermore, the level of ACC phosphorylation was comparable to controls at the basal level. Please see Supplementary Fig. 17A, Page 11, 12. We also tested AMPK activity in Dex administered SIRT6 inhibitor-treated mice. Here, we found that phosphorylation of ACC in Dex administered SIRT6 inhibitor-treated mice was comparable to control Dex treated mice. However, phosphorylation of ACC was low in SIRT6 inhibitor-treated mice at the basal level. Please note that we could not measure p-AMPK levels in Dex or Veh-administered SIRT6 inhibitor-treated mice. Please see, Supplementary Fig. 21H, Page 15.

As per the suggestion, we tested whether AMPK inhibitor compound C could rescue reduction in SIRT6 overexpressing primary myotube diameter. Interestingly, we found that AMPK inhibitor compound C could not rescue SIRT6 overexpression-induced reduction in myotube diameter. In addition, inhibition of AMPK spontaneously led to a reduction in myotube diameter. These results suggest that long-term deficiency of SIRT6 might be responsible for changes in AMPK signaling in msSIRT6-KO mice muscle. Therefore, we speculate that AMPK signaling might not be directly involved in mediating the regulatory action of SIRT6 on muscle. Please see Supplementary Figure 17B-17D, Page 11, 12.

Comment 3: It is important to note that in reference 51 (on Akt 1 and 2 KO mice) that AMPK activity was also impacted. As such, the discussion on page 15 related to Akt1 and Akt2 should also include discussion about AMPK, such that "SIRT6 controls the activity of PI3K/AKT signaling while protecting against Dex induced muscle mass loss through the combined action of AKT1/AKT2" and potentially AMPK.

Response: We thank the reviewer for highlighting an interesting effect of prolonged AKT1/2 deficiency on AMPK. Our in vitro experiment using AMPK inhibitor in SIRT6 overexpressing myotubes suggests that AMPK might not be playing a key role in protection against Dex-induced muscle mass loss. Therefore, we have not included any further discussion on AMPK in the revised manuscript. Please see Supplementary Figure 17B-17D, Page 11, 12.

Comment 4: Although not critical for a resubmission, a nice addition to the paper would be a more robust measure of AMPK activity, be it pACC or an activity assay.

Response: We thank the reviewer for the suggestion. We analysed the change in AMPK activity by testing ACC phosphorylation, an AMPK downstream target in Dex treated msSIRT6-KO. While Dex-treated control mice showed increased ACC phosphorylation, msSIRT6-KO mice were resistant against

Dex-induced increase in ACC phosphorylation. Furthermore, the level of ACC phosphorylation was comparable to controls at the basal level. Please see Supplementary Fig. 17A, Page 11, 12. We also tested AMPK activity in Dex administered SIRT6 inhibitor-treated mice. We found that phosphorylation of ACC in Dex administered SIRT6 inhibitor-treated mice was comparable to control Dex treated mice. However, phosphorylation of ACC was low in SIRT6 inhibitor-treated mice at the basal level. Please see, Supplementary Fig. 21H, Page 15.

Comment 5: The title and abstract is misleading as it infers (at least to the Reviewer/reader) that SIRT6 is "directly" modulating glucocorticoid-induced skeletal muscle atrophy. It is clear that these effects are secondary to a regulatory axis that includes AKT1/2 and potentially AMPK. The title and abstract, and conclusions in general, should more accurately reflect the data and mechanism of action.

Response: We thank the reviewer for the thoughtful comment. We have revised the manuscript as per your suggestion.

Comment 6: Given that hierarchically speaking, SIRT6 has many pathways that it regulates, in many different tissues, to suggest targeting SIRT6 as a potential strategy for treating muscle wasting associated with stress and disease conditions seems premature and potentially problematic. How would you do so specifically in skeletal muscle so as to avert off target effects/consequences in other tissues? Also, why not modulate Akt (or AMPK), given that it works through such a mechanism? The discussion of modulating SIRT6 as a means of treating muscle wasting should be balanced by potential issues with such an approach, given the broad contexts of SIRT6 action.

Response: We thank the reviewer for the insightful comment. We have made suggested changes in the manuscript considering the off-target effect of the SIRT6 inhibitor on multiple tissues considering its role in a plethora of functions. Therefore, we have also included the possibility of testing AKT inhibitors as a potential therapeutic strategy against stress-induced muscle wasting and the potential issues with the usage of SIRT6 inhibitors in the discussion section of the revised manuscript. Please see Page 17, 18.

Comment 7: Related to the ponceau stains for the puromycin blots. Please provide quantification of the ponceau stains. Also, it is unclear if the SUnSET analysis corrected for the ponceau stained membranes? For those SUnSET membranes that a ponceau stains exist, the puromycin should be corrected for ponceau. It is also unclear if the SUnSET is corrected for GAPDH/loading control? Please state in Methods.

Response: As per suggestion, we have included puromycin quantification normalised to ponceau, and also ponceau quantification. In our previous revised manuscript, the puromycin levels were normalised with respect to GAPDH. We have updated our figures, figure legends, and method sections as per suggestion. Please see Supplementary fig. 16J-16L, 21A, 21B. Please see Page 32.

Comment 8: While the unexpected recombination events is a concern, a major concern with the germline approach, which is not stated, is secondary (or tertiary etc.) changes/adaptations that occur in the muscle as a result of long-term loss of SIRT6. Reference 51 is a perfect example of such a problem in that loss of Akt1/2 resulted in AMPK activation. This can result in misinterpretation of the role of

the protein that is, for example, knocked out. This point about secondary etc. changes should also be incorporated into the relevant paragraph and tied back to the author's results.

Response: As per suggestion, we have included these points in the discussion section. Please see Page 18.

Comment 9: The manuscript is difficult to read in places. A robust review for syntax and grammar is recommended. One example is the last sentence of paragraph 2 of the Discussion.

Response: As per suggestion, we have revised our manuscript using

Grammarly. **Reviewer #3**

The present version of the study by Mishra and collaborators is markedly improved in comparison to the previous one. The authors satisfactorily addressed all the concerns raised in the first revision round. However, there is still one single issue that needs to be clarified, referred to author comment to point n. 4. In this regard, it is unclear if the hypothesis provided by the authors to discuss the differences occurring between molecular and phenotype discrepancies are reported somewhere in the paper. If not, they should...

Response: We thank the reviewer for the suggestion to improve the manuscript. As per suggestion, we included the representative image showing increase in size in COX positive myofibers. Moreover, we have also added our hypothesis to discuss the discrepancies observed between molecular and phenotypic levels in the discussion section. Please see Page 18, 19.

Reviewers' Comments:

Reviewer #2:

Remarks to the Author:

Substantial and commendable changes have been made.

My main concern revolves around statements related to therapeutic targets.

1. Abstract. The authors state "SIRT6 inhibition as a potential strategy for treating muscle wasting associated with stress and disease conditions.". Per comment 6 in my previous review this is highly problematic and should be amended. Other parts of the manuscript appear to have addressed this by being more open to the issues of SIRT6 modulation as a therapeutic.

2. Line 651. Any statement/reference to the use of Akt activators should be removed. The authors are encouraged to better understand the difficulties of modulating Akt (PMID: 17057754) especially as it relates to its clear oncogenic actions, and the dangers of using an Akt activator.

Big picture, statements related to the therapeutic actions of this work should be more general.

1 **Reviewer #2**

2 Substantial and commendable changes have been made. My main concern revolves around statements
3 related to therapeutic targets.

4 Response: We thank the reviewer for appreciating our efforts in revising the manuscript. Furthermore,
5 we thank the reviewer for providing insightful suggestions that has improved our manuscript. Please
6 find the response to the reviewer comment below.

7 Comment 1: Abstract. The authors state "SIRT6 inhibition as a potential strategy for treating muscle
8 wasting associated with stress and disease conditions.". Per comment 6 in my previous review this is
9 highly problematic and should be amended. Other parts of the manuscript appear to have addressed this
10 by being more open to the issues of SIRT6 modulation as a therapeutic.

11 Response: We have now removed any problematic statement from the abstract and revised it as per
12 suggestion.

13 Comment 2: Line 651. Any statement/reference to the use of Akt activators should be removed. The
14 authors are encouraged to better understand the difficulties of modulating Akt (PMID: 17057754)
15 especially as it relates to its clear oncogenic actions, and the dangers of using an Akt activator.

16 Response: We thank the reviewer for providing the details of the off-target action of modulating AKT
17 signaling. We have now removed all the statements that pointed out towards the usage of AKT
18 activators for therapeutic purpose. (Line 601, 651 in previous version of the manuscript)

19 Comment 3: Big picture, statements related to the therapeutic actions of this work should be more
20 general.

21 Response: We have revised the manuscript as per the suggestion.

22

23

24